# Hard rocks and deep wetlands beneath Thwaites Glacier in Antarctica

Ole Zeising [1] ✉, Olaf Eisen [1,2], Coen Hofstede [1], Ronan Agnew [3], Alex Brisbourne [3], Andrew O. Hoffman [4,5] & Sridhar Anandakrishnan [6], On behalf of the GHOST Team

Thwaites Glacier in West Antarctica is losing ice rapidly and is considered especially vulnerable to retreat, but predictions of its future remain limited by uncertainties about its subglacial properties. Here we show results from 344 km of vibroseismic surveys collected along and across the glacier. The data reveal a heterogeneous bed of elevated ridges with steep upstream-facing slopes that form crag-and-tail landforms resisting fast flow. Between these ridges lie basins filled with consolidated sediments. Subglacial water is widespread, occurring in bed depressions and on topographic highs, including an active lake composed of tens of metres of highly porous, water-saturated sediments. Across the glacier, the bed beneath the eastern margin is mostly hard but contains isolated pockets of softer material. These findings demonstrate current models do not capture the full complexity of the bed beneath Thwaites Glacier, where water-bearing sediments and steep basal slopes strongly affect ice flow and retreat.

The West Antarctic Ice Sheet is experiencing substantial mass loss[1,2], making it a major contributor to global sea-level rise[3]. In the Amundsen Sea Embayment, the primary driver of mass loss is the intrusion of warm Circumpolar Deep Water into the cavities of ice shelves[4,5]. The warm water leads to enhanced melt rates at the steep ice base near the grounding line[6,7], resulting in thinning of the ice shelves[1] and a reduction in their buttressing effect[8–12]. Consequently, outlet glaciers such as Pine Island and Thwaites experience accelerated ice discharge into the ocean[10]. Since 2002, the Thwaites drainage basin has lost mass at an average rate of $60.5 \pm 9.2$ Gt a$^{-1}$, corresponding to a global mean sea-level rise of $0.17 \pm 0.03$ mm a$^{-1}$ (or 5%), making it the largest Antarctic contributor to sea-level rise[13,14].

Thwaites Glacier is grounded below sea-level on a retrograde slope that deepens inland[15], making it susceptible to the Marine Ice Sheet Instability. This makes Thwaites Glacier of particular concern in future warming scenarios because of its potential for rapid retreat and subsequent destabilization of substantial portions of the West Antarctic Ice Sheet[16]. Ice loss from Thwaites Glacier is anticipated to persist[17,18], with severe consequences for future sea-level rise, though model projections vary widely, depending on the physical processes included in the model, forcing applied to the system, and the boundary conditions. Understanding the factors that lead to the present mass loss and the processes that govern future developments is crucial for improving projections of its evolution.

Thwaites Glacier, ~120 km wide at the grounding line, has two floating ice shelves: the eastern ice shelf and the western ice tongue. In 2009, the Thwaites Western Ice Tongue detached from its pinning point and subsequently transitioned into a fragmented assemblage of icebergs. This event led to further acceleration in flow at the grounding line[10,19,20]. Simulations suggest that Thwaites Western Ice Tongue now provides almost no buttressing[21]. The loss of buttressing resulted in dynamic thinning of the grounded ice upstream of the grounding line[22,23]. Both dynamic thinning and intense basal melting at the grounding line have led to a grounding line retreat of up to 0.8 km a$^{-1}$ [6,7,22,24,25].

Numerical models project that Thwaites Glacier will continue to accelerate and thin, and the grounding line will retreat[17,18,26,27], yet substantial uncertainties persist—mainly due to poorly constrained bed properties that strongly influence ice dynamics. Accurate ice-sheet modelling requires well-resolved ocean thermal forcing, as well as onshore basal topography, roughness, and basal mechanical properties[8,17,26,28]. Bed topography plays a key role in grounding-line stability: retrograde slopes promote rapid retreat, while topographic highs delay it. Simulations show that two ridges—Ghost Ridge (~60 km upstream of the grounding line) and Lower Thwaites Ridge (LTR, ~130 km upstream)—can slow retreat, but once the grounding line passes them, rapid collapse of the West Antarctic Ice Sheet becomes possible[26]. The mechanical behaviour of the ice-bed contact further

---

[1]Glaciology, Alfred Wegener Institute Helmholtz Centre for Polar and Marine Research, Bremerhaven, Germany. [2]Faculty of Geosciences, University of Bremen, Bremen, Germany. [3]Ice Dynamics and Palaeoclimate, British Antarctic Survey, Natural Environment Research Council, Cambridge, UK. [4]Department of Earth, Environmental, and Planetary Science, Rice University, Houston, TX, USA. [5]Lamont-Doherty Earth Observatory, Columbia University, Palisades, NY, USA. [6]Department of Geosciences, and Earth and Environmental Systems Institute, Pennsylvania State University, University Park, PA, USA. A list of members and their affiliations appears in the Supplementary Information. ✉e-mail: ole.zeising@awi.de

modulates retreat rates by controlling basal sliding. Hard beds generate high basal drag, restricting ice flow. Under such conditions, high melt rates at the ice–ocean interface primarily cause localized thinning, while still enabling rapid grounding-line retreat. In contrast, soft beds of dilatant till or unconsolidated sediments produce low basal drag, enhancing basal sliding and glacier acceleration. Although this can initially delay grounding-line retreat, the associated upstream thinning may accelerate the retreat later on[8,26]. Field observations are crucial for mitigating uncertainties in modelling. The subglacial properties of Thwaites Glacier have been investigated through a combination of remote sensing observations and modelling, which revealed a hard bed within 80 km of the grounding line as well as sedimentary basins in depressions between bands of higher bed topography[29,30]. Airborne swath radar surveys, combined with active seismic measurements, have to some extent provided the bed topography and mechanical properties beneath Thwaites Glacier[31–36]. These datasets reveal widespread elongated subglacial landforms, including crag-and-tails and mega-scale glacial lineations (MSGLs).

The subglacial hydrological system plays a crucial role in modulating basal friction and melting, yet its detailed structure and influence remain poorly constrained. Satellite altimetry methods have revealed active subglacial lakes beneath Thwaites Glacier, showing periodic filling and drainage events[37–39]. Drainage episodes caused transient ice-flow accelerations of less than 3%[38], suggesting that the active subglacial hydrological system has limited direct effect on the current thinning and retreat of Thwaites Glacier. Numerical models of subglacial hydrology suggest that these lakes are connected by channels and that some channels extend >170 km to the grounding line[40]. However, simulation results strongly depend on frictional heating, geothermal heat flux, and bed topography, which can be determined by geophysical measurements. Despite previous seismic and radar surveys, the subglacial environment beneath Thwaites Glacier remains poorly defined. Key bed properties, including mechanical properties, roughness, and hydrological characteristics, are major controls on ice flow that remain insufficiently constrained by direct observations.

Here, we present 344 km of vibroseismic profiles collected along and across Thwaites Glacier. This dataset provides substantially greater spatial coverage and a higher signal-to-noise ratio than previous studies, while maintaining high horizontal resolution. We combine amplitude, velocity, and structure analysis of seismic reflections to characterize basal topography, subglacial materials, and hydrological features, thereby resolving basal heterogeneity at scales not previously observed at Thwaites Glacier or elsewhere underneath grounded ice in Antarctica. By evaluating acoustic impedance and interpreting it in terms of spatially variable basal drag, our results provide critical constraints for ice sheet models. Such constraints are particularly important for improving projections of future mass loss from one of Antarctica's most vulnerable outlet glaciers, where basal conditions strongly influence ice acceleration and grounding-line retreat.

## Results

### Vibroseismic profiles in the context of Thwaites Glacier

As part of the ITGC-GHOST project we collected a 210 km profile in summer 2022–23 (hereafter *along-flow profile*) and a 134 km profile in summer 2023–24 (hereafter *across-flow profile*) on Thwaites Glacier (Methods). To place these profiles in the context of Thwaites Glacier, we utilise MEaSUREs ice flow velocities[41,42] and geometry from Bedmap3[43,44] (Supplementary Movie 1). The along-flow profile spans roughly half the glacier's length, starting 72 km upstream of the grounding line and extending to 282 km upglacier (Fig. 1a,b,e). Along the profile, the glacier accelerates from 90 to 460 m a$^{-1}$, increasing further downstream to 3 km a$^{-1}$ in the heavily crevassed region near the grounding line. From the grounding line to GHOST Ridge (km 53), the bed lies at an elevation of ~−1000 m, rising to −840 m at GHOST Ridge. Upstream, the retrograde bed deepens to an average elevation of −1300 m, with the lowest point at −1500 m. Ice thickness increases from ~900 m at the grounding line to 2150 m within 72 km, exceeding 2600 m at the upstream end of the along-flow profile.

The across-flow profile is located 123 km upstream of the grounding line, where the glacier is 190 km wide. The seismic profile extends eastwards from the centre of the main trunk (beginning 5 km west of the along-flow profile) beyond the eastern shear margin (Fig. 1c,d,e). Ice flow velocities in the main trunk exceed 300 m a$^{-1}$, decreasing rapidly toward the western shear margin but more gradually eastward. Outside Thwaites Glacier, ice flow velocities drop below 10 m a$^{-1}$. The main trunk lies on the deepest bed (<−1000 m). In the west, the bed elevation increases rapidly to +100 m at the shear margin; to the east, it rises abruptly over 10 km from −1300 m to −750 m.

### Bed properties along ice-flow direction

The vibroseismic along-flow profile and its basal property analysis are presented in Fig. 2. For reference, the profile is divided into eight segments (I–VIII, downstream to upstream). Enlarged views of regions III and IV are shown in Fig. 3, with additional enlargements in Supplementary Figs. 1–5. In the following, we first provide an overview of the regions exhibiting rough terrain on a larger scale along the profile, then describe small-scale variations and material properties from the amplitude and velocity analysis. Bed topography alternates between smooth, gently sloping sections (II, IV, V, VII) and rough, steeply sloping areas with pronounced elevation changes (I, III, VI, VIII).

At the downstream end of the seismic profile, the bed exhibits such a rugged terrain (region I: km 72 to 90). The largest elevation change in this region is a 150 m high flank, with a steep basal slope of 33% that faces in the upstream direction. The two largest regions of elevated and rough topography are LTR and Upper Thwaites Ridge, located at km ~115–145 (III) and km ~205–240 (VI) upstream from the grounding line. LTR has the highest elevation of −1050 m, surpassing its surroundings by approximately 350 to 400 m. The ridge comprises multiple elevations and depressions separated by steep basal slopes reaching up to ±27%. The topography of the UTR is less pronounced in the seismic profile compared to the LTR, primarily due to the profile's positioning. The ridge elevation is substantially higher to both the east and west of the seismic profile. At the upstream end of the profile (VIII: km 260 to 282), the bed is less rough than at the ridges, yet still exhibits steep flanks in the direction of ice flow.

At both ridges and in the downstream region I, we identified steep upstream-facing slopes (stoss sides), which exhibit acoustic impedance values consistent with consolidated material, possibly bedrock (Fig. 3). In contrast, the lee sides are gently sloping and composed of dilatant till or unconsolidated sediments at the ice–bed interface. Several of these features are probably crag-and-tail landforms comprising a steep, approximately 100 m high stoss side (crag), followed by a lee-side tail extending for several kilometres. Tails are generally flat or slightly declining, with slopes up to 10%, and drape downwards over the underlying bed at the downstream side. We observed moats up to ~120 m deep upstream of crags[45,46] and similar features but likely with a different formation mechanism downstream of tails. These moats are narrow, elongated depressions, formed by focused basal water flow or enhanced erosion adjacent to harder bed obstacles. Reflection coefficients and acoustic impedance values suggest that the tails are overlain by dilatant till. The analysis of interval velocities of deeper regions (2450–2900 m s$^{-1}$) indicates the presence of consolidated material beneath the till. At LTR (III), we identified three prominent crag-and-tail features with a tail length between 1.5 and 4 km (Supplementary Fig. 3). At UTR (VI), such features appear less developed, possibly due to the seismic profile's position between two higher-elevation regions, to the east and west, where more pronounced structures may exist.

Downstream of the LTR lies a valley (II), ~20 km wide and 300 m deep. Along the gently sloping flanks are small sedimentary basins, each with a thickness of a few tens of meters overlying a deeper structure that occasionally comes to the surface. The small basins have parallel stratification, whereas the underlying structure at some places shows oblique stratification (Supplementary Fig. 1). The acoustic impedance indicates that these basins are filled with dilatant till and unconsolidated sediments

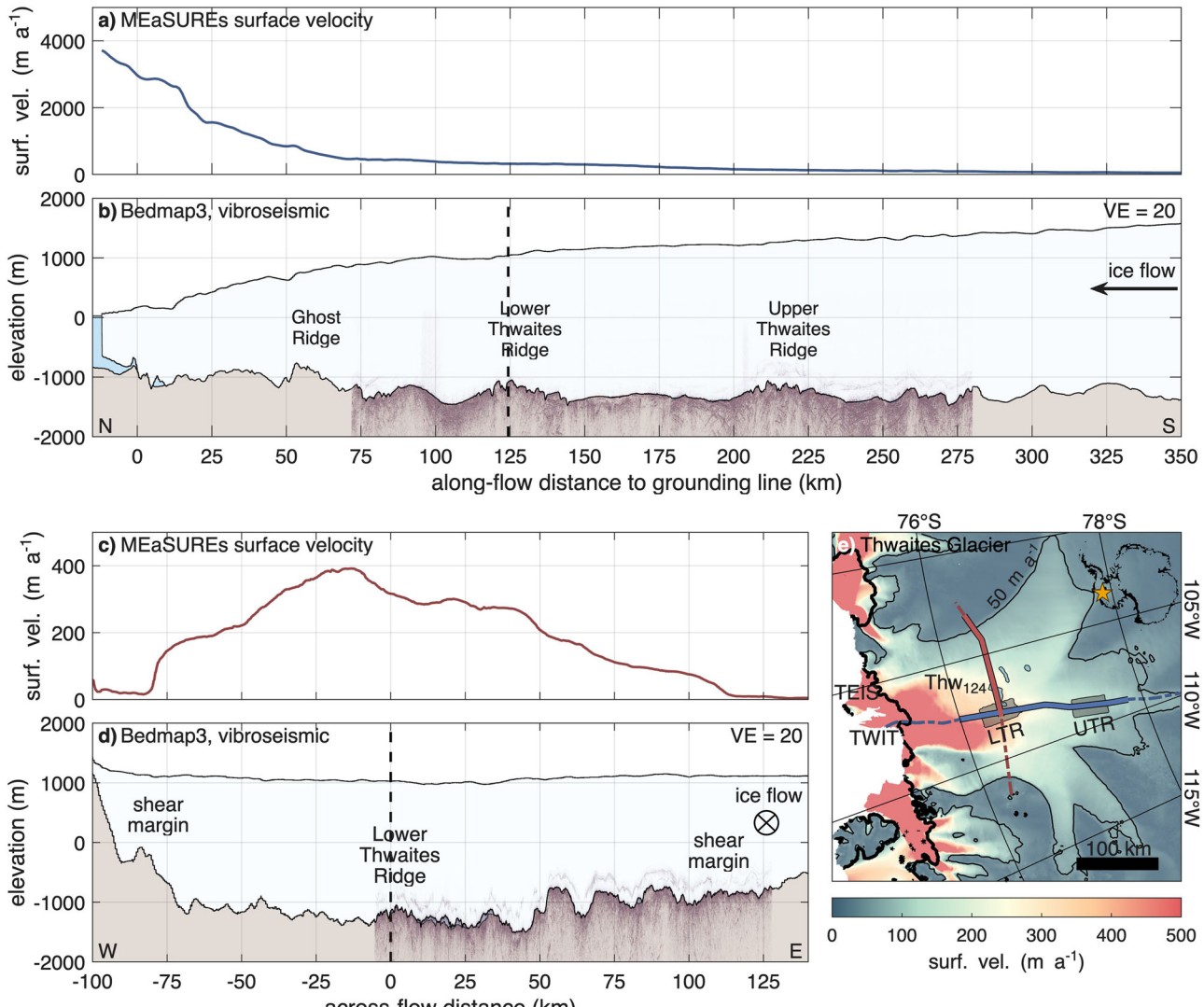

**Fig. 1 | Vibroseismic profiles in context of Thwaites Glacier. a, c** MEaSUREs ice flow velocity[41,42] and **b, d** Bedmap3 geometry[43,44] with vibroseismic profiles in along-flow (**a, b**) and across-flow direction (**c, d**). Elevation relative WGS 84. The sections are vertically exaggerated by a factor of VE = 20. Dashed lines indicate the profile intersection. **e** Map of Thwaites Glacier showing the along-flow profile (blue), the across-flow profile (red), and their extensions (dashed-lines). The background shows surface flow velocity from MEaSUREs[41,42], with the 50 m a[−1] contour line. Also shown are the grounding line (thick black line), subglacial lakes Thw$_{124}$, Thw$_{142}$, and Thw$_{170}$ (blue areas), swath airborne radar data at Lower Thwaites Ridge (LTR) and Upper Thwaites Ridge (UTR, grey areas)[35], as well as the two floating extensions Thwaites Eastern Ice Shelf (TEIS) and Thwaites Western Ice Tongue (TWIT). The inset map in the upper-right corner locates Thwaites Glacier within Antarctica, with its position marked by the orange star.

with consolidated or lithified patches in between. At the deepest part of the basin, seismic reflections extend down to a thickness of 500 m. However, off-nadir reflections from nearby MSGLs cannot be entirely dismissed.

In between the two ridges is a predominantly flat area, ~50 km wide, with basal slopes of less than 5% (IV and V). Upstream of LTR, the seismic measurements reveal a 20 km wide and up to 300 m deep basin (IV, Fig. 3). The derived acoustic impedance as well as the p-wave velocities of $2680 \pm 440$ m s[−1] indicate that the basin is likely filled with consolidated sediments. Region V, located between the basin and UTR, exhibits a range of basal conditions, from patches of water to consolidated sediments.

At the upstream area of UTR lies a flat ice–bed interface (VII) spanning 20 km, which has previously been identified as a MSGL[35]. In the downstream region, the acoustic impedance suggests the presence of water within the sediment. Conversely, in the slightly elevated upstream area, dilatant till and unconsolidated sediments prevail. Crossing reflections beneath the ice–bed interface indicate the appearance of off-nadir reflections from parallel MSGLs.

## Bed properties across ice-flow direction

The across-flow profile is divided into five regions, labelled with Roman numerals from IX (west) to XIII (east). Figure 4 shows the complete profile, while Figs. 5 and 6 present enlarged views of key areas, including regions IX and X (subglacial lake) and region XIII (shear margin). Additional close-ups are provided in Supplementary Figs. 6–8.

At the westernmost end of the across-flow profile, in the central trunk of Thwaites Glacier, lies LTR (region IX, Fig. 4 and Supplementary Fig. 6). Here, the bed is rough, ranging from $-1270$ to $-1060$ m. The basal reflection coefficient indicates predominantly hard, consolidated sediments, interrupted by two areas consistent with unconsolidated, possibly dilatant sediments. One occurs where the across-flow profile intersects the along-flow vibroseismic line, at the downstream end of a crag-and-tail feature. The second location lies on the eastern side of LTR, at the downstream edge of a small, elevated area.

Within region X (Fig. 4 and Supplementary Fig. 6), we observe consistent indicators of basal water-sediment mixtures beneath the ice (km 8 to 40). These include predominantly negative reflection coefficients and lower

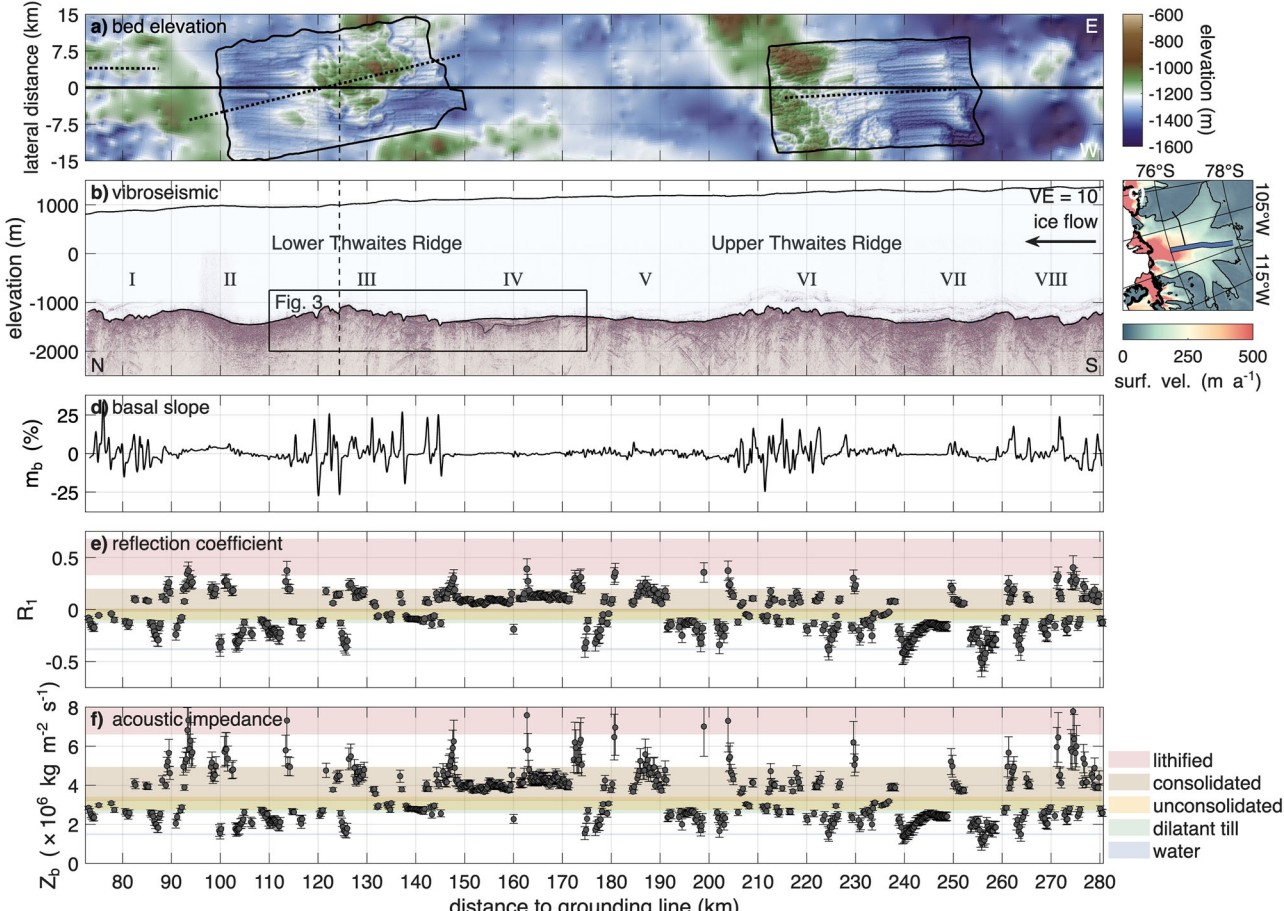

**Fig. 2 | Along-flow vibroseismic profile in context of lateral bed elevation and derived parameters. a** Map view of bed elevation (WGS 84) from Bedmap3[43,44] and swath airborne radar (black outline[35]) in lateral direction. The dotted lines represent seismic surveys at LTR[34], UTG[32,33], and an unpublished seismic reflection survey from summer 2023–24 in region I[75]. **b** Kirchhoff migrated vibroseismic section with a vertical exaggeration VE = 10. The Roman numerals label different areas of the bed. The dashed line indicates the intersection with the across-flow profile. **c** Map of Thwaites Glacier with location of vibroseismic section (blue line). Details are presented in Fig. 1c. **d** Basal slope $m_b$ in ice flow direction, calculated from a mean gradient over twice the Fresnel zone radius. **e** Basal reflection coefficient $R_1$. **f** Basal acoustic impedance $Z_b$. Coloured areas in **d** and **e** refer to sediment classification, as indicated in the legend below the panels. The shown profiles are a function of distance to grounding line. Ice flow from right to left. Error bars indicate uncertainties in the reflection coefficient and acoustic impedance, as described in the Methods.

acoustic impedance values than expected for dilatant till, pointing to a soft basal layer with varying porosity and/or effective pressure, possibly with the presence of free water.

Across this region, an active subglacial lake (Thw$_{124}$) with a surface extent of approximately 257 km² is evident from time-variable surface elevation (Fig. 5)[38]. From 2019 to the beginning of 2025, the surface lowered by 8.8 m on average, corresponding to a drainage volume of 2.26 km³. The migrated seismic section reveals large, acoustically transparent zones without substantial reflections between two strong reflections at the ice base. We interpret the first reflection as the base of the ice, which is consistent with the ice thickness derived from radar data[47]. Interval velocity analysis of the transparent zones yields p-wave velocities ranging from 1540 to 2380 m s⁻¹. Velocities at the lower end of this range are indicative of highly porous, water-saturated dilatant sediments. From the mean interval velocity, we derive an average thickness of these zones to be ~70 ± 15 m, with a maximum thickness of 160 ± 34 m. These transparent zones are located downstream of elevated regions, span up to 6 km in width and cover a major portion of the subglacial lake area (Fig. 5). A 100 m deep and 500 m wide channel separates two of these zones in the western part near the LTR. The lateral boundaries of these water-containing zones exhibit steep slopes, whereas the second strong reflection indicates gentler slopes. Outside the transparent zones, the reflection coefficient at the ice base indicates the

presence of unconsolidated sediment with a higher acoustic impedance than within the transparent zones, suggesting that the water-containing sediments are resting on top of a harder bed. This interface between the highly porous sediments and the harder bed causes the second strong reflection with a median bed elevation of −1350 m. At the eastern margin of the region X (km 35 to 45), the bed further decreases to the lowest point of −1500 m. Within this area, a sedimentary basin of unconsolidated sediments spanning 3 km exists. At both sides of the basin, a hard bed of consolidated sediments rises by a few tens of meters.

In the region XI further to the east (Fig. 4 and Supplementary Fig. 7), the bed rises by 750 m within 7 km and forms a subglacial mountain with a maximum elevation of −750 m at km 45 to 62. On the western flank of this mountain lies a 3 km wide plateau. The basal reflection coefficient along the valley-facing side of this plateau (km 48) indicates water at the ice–bed interface, while the mountain-facing side (km 51) consists of unconsolidated material. The acoustic impedance at the 5 km wide top of the mountain suggests the presence of consolidated sediments as well as lithified material. However, at one location on the summit, in a 1 km wide area that is a few tens of meters deeper, the migrated seismic section shows a transparent zone with strongly negative reflection coefficients, indicating the presence of water (km 57). The summit of the mountain is separated from the plateau in the east (regions XII and XIII) by a 500 m deep and 7 km wide

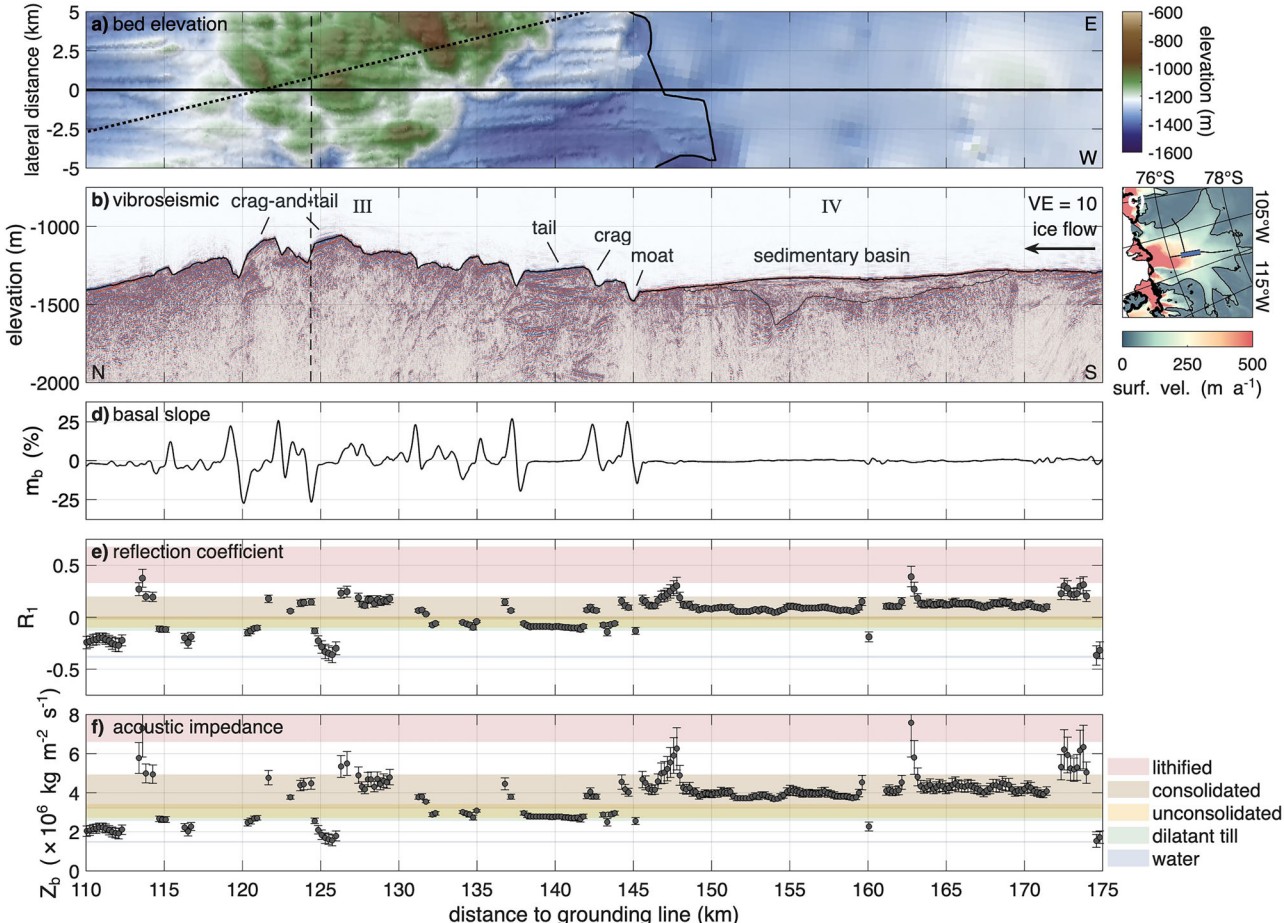

**Fig. 3 | Region III and IV of the along-flow profile. a** Map view of bed elevation (WGS 84) from Bedmap3[43,44] and swath airborne radar (black outline[35]) in lateral direction. The dotted line represents a seismic survey at LTR[34]. **b** Kirchhoff migrated vibroseismic section with a vertical exaggeration VE = 10. The dashed line indicates the intersection with the across-flow profile. **c** Map of Thwaites Glacier with location of vibroseismic section (blue line). Details are presented in Fig. 1c. **d** Basal slope $m_b$ in ice flow direction, calculated from a mean gradient over twice the Fresnel zone radius. **e** Basal reflection coefficient $R_1$. **f** Basal acoustic impedance $Z_b$. The shown profiles are a function of distance to grounding line. Ice flow from right to left. Error bars indicate uncertainties in the reflection coefficient and acoustic impedance, as described in the Methods.

valley. The ice–bed interface within the valley is heterogenous, with a wide range of reflection coefficients indicating the presence of water, dilatant till and consolidated sediments as well as lithified materials.

The plateau that extends eastward beyond the eastern shear margin ranges in elevation between −1100 and −720 m (Fig. 4 and Supplementary Fig. 7). The reflection coefficient at the ice–bed interface in the region between the valley in the west and the shear margin in the east (XII, km 70 to 105) shows jumps between consolidated sediments and water-containing sediments, but nothing in between, similar to the region XI in the west. Some areas show signs of water at the ice base. At one of these locations, the migrated seismic section shows a 100 m thick and 1.5 km wide transparent area. Other areas in this region show the presence of dilatant till and consolidated or lithified sediments at the base of the ice. Specifically, consolidated and lithified sediments are more prevalent in higher-elevation regions, while dilatant till is more common at lower elevations. It is noteworthy that areas indicating water at the ice base are found in both higher and lower-elevated beds.

The eastern shear margin of Thwaites Glacier is located approximately between 108 and 115 km west of the along-ice flow profile (Fig. 6). Within this zone, ice flow velocity decreases from 50 to 10 m a$^{-1}$. The vibroseismic data reveal a relatively flat bed that varies between −780 and −880 m. On the fast-flowing side of the shear margin, high reflection coefficients at the ice-bed interface suggest a predominantly hard bed of consolidated or

lithified material. However, isolated zones of water-bearing dilatant till are also present. On the slow-flowing side, the bed is largely composed of consolidated sediments. Farther outside of Thwaites Glacier (km 115 to 127), the bed remains flat and consists primarily of consolidated and lithified material.

## Englacial reflections

The two vibroseismic profiles exhibit a band of strong englacial reflections located at approximately 200 to 450 m above the ice base that generally follow the bed topography. The along-ice flow profile (Fig. 2b) reveals englacial reflections along its entire length, with intensity varying according to bed morphology. Reflections are generally strongest over rough, elevated bed sections, particularly from upstream of UTR to its downstream end (regions VIII to VI). They weaken over the flat terrain of regions V and IV, strengthen slightly in regions III and II, and intensify again in region I.

In the across-flow profile (Fig. 4b), englacial reflections are present throughout. However, the topography of the englacial reflections subtly differs from the bed topography. The englacial reflections are particularly pronounced on the fast-flowing side of the shear margin but almost disappear within the shear margin. Outside the shear margin, they reappear, but this time also within the lower 50% of the ice thickness. Englacial reflections may extend to greater heights, but they were obscured by the direct and diving wave.

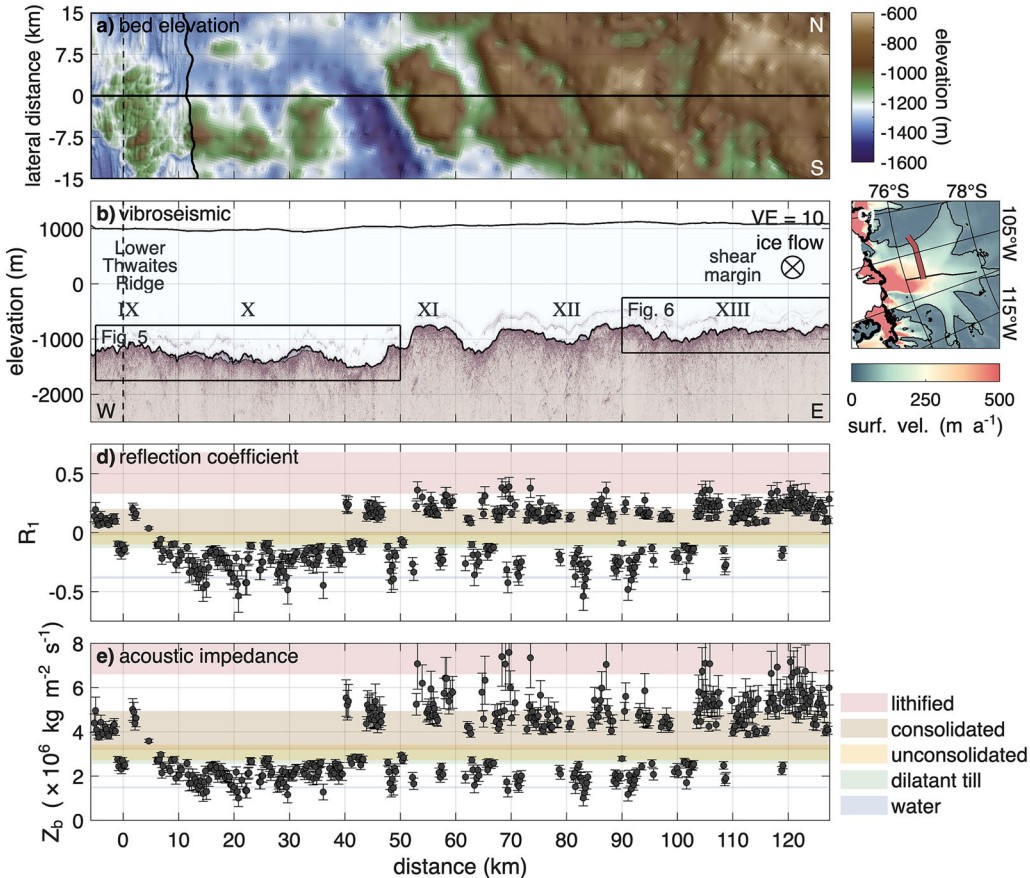

**Fig. 4 | Across-flow vibroseismic profile in context of lateral bed elevation and derived parameters. a** Map view of bed elevation (WGS 84) from Bedmap3[43,44] and swath airborne radar (black outline[35]) in lateral direction. The white, dotted line marks the outline of a subglacial lake, derived from the drainage between 2022 and 2024. **b** Kirchhoff migrated vibroseismic section with a vertical exaggeration VE = 10. The dashed line indicates the intersection with the along-flow profile. **c** Map of Thwaites Glacier with location of vibroseismic section (red line). Details are presented in Fig. 1c. **d** Basal reflection coefficient $R_1$. **e** Basal acoustic impedance $Z_b$. The shown profiles are a function of distance to the along-flow profile. Ice flow into the page. Error bars indicate uncertainties in the reflection coefficient and acoustic impedance, as described in the Methods.

## Discussion

The future development of Thwaites Glacier will mainly be determined by the properties of the bed beneath the glacier, the magnitude of ocean thermal forcing, and, to a lesser degree, the surface mass balance and properties of the ice. Bed characteristics, including the basal topography, basal roughness, mechanical properties and subglacial hydrology, control the basal drag and will have an impact on the future grounding line retreat, acceleration and inland thinning.

### Bed topography

The vibroseismic profile in the along-flow direction provides improved constraints on bed topography and basal roughness relative to existing regional datasets, revealing pronounced large-scale variability. In elevated regions, such as LTR and UTR, the bed is rough and steeply sloping. These elevated ridges play a key role in regulating ice sheet dynamics of Thwaites Glacier. Simulations demonstrate that the future grounding line retreat will be slowed over the ridges, particularly GHOST Ridge and LTR, independent of the mechanical properties of the ridges[26]. In these simulations, when the grounding line passes across the LTR, the retreat accelerates over the regions of lower elevation, whilst UTR does not induce any slowing.

The reliability of these simulations depends on the extent to which the bed topography products used in the models accurately represent the actual topography. We compared our vibroseismic bed topography with the latest radar-based topography products, Bedmap3[43,44] and BedMachine Antarctica v3[15,48] (Supplementary Figs. 9, 10). The comparison shows good agreement with Bedmap3, which captures the high basal roughness and

steep slopes observed in our data. In contrast, BedMachine presents a smoother profile with less pronounced slopes. Notably, while the vibroseismic data show similar features at LTR and UTR, BedMachine underestimates UTR's elevation and smooths out topographic peaks. These discrepancies are even greater in older models like Bedmap2[49]. Further simulations will be necessary to assess the impact of newer and more precise bed topography products like Bedmap3 on the simulated grounding line retreat.

### Basal mechanical properties

While the bed topography plays a key role in the retreat of the grounding line, the mechanical properties of the bed also influence the glacier flow velocity through basal drag. Ice-sheet models, utilised to simulate the future evolution of Thwaites Glacier, are often based on the inversion of the basal drag. Basal drag estimates for West Antarctica have been presented from an L-curve analysis of inverse modelling utilising subglacial hydrology simulations and assuming a uniform bed[50]. High basal drag was found at the ridges—particularly at GHOST Ridge and LTR—while basal drag at UTR is lower (Supplementary Figs. 9, 10). The sensitivity of grounding line retreat to the presumed basal mechanical properties has also been highlighted[26]. In these simulations, a uniform hard bed restricts ice acceleration and causes rapid retreat under ocean forcing, whereas a uniform soft bed allows faster flow but delays retreat due to inland thinning. At Thwaites Glacier, our seismic measurements and derived acoustic impedance reveal a mixed bed, comprising approximately 50% hard bed, 43% soft bed, and 7% intermediate bed (Fig. 7, Supplementary Fig. 11). In the higher-elevation regions,

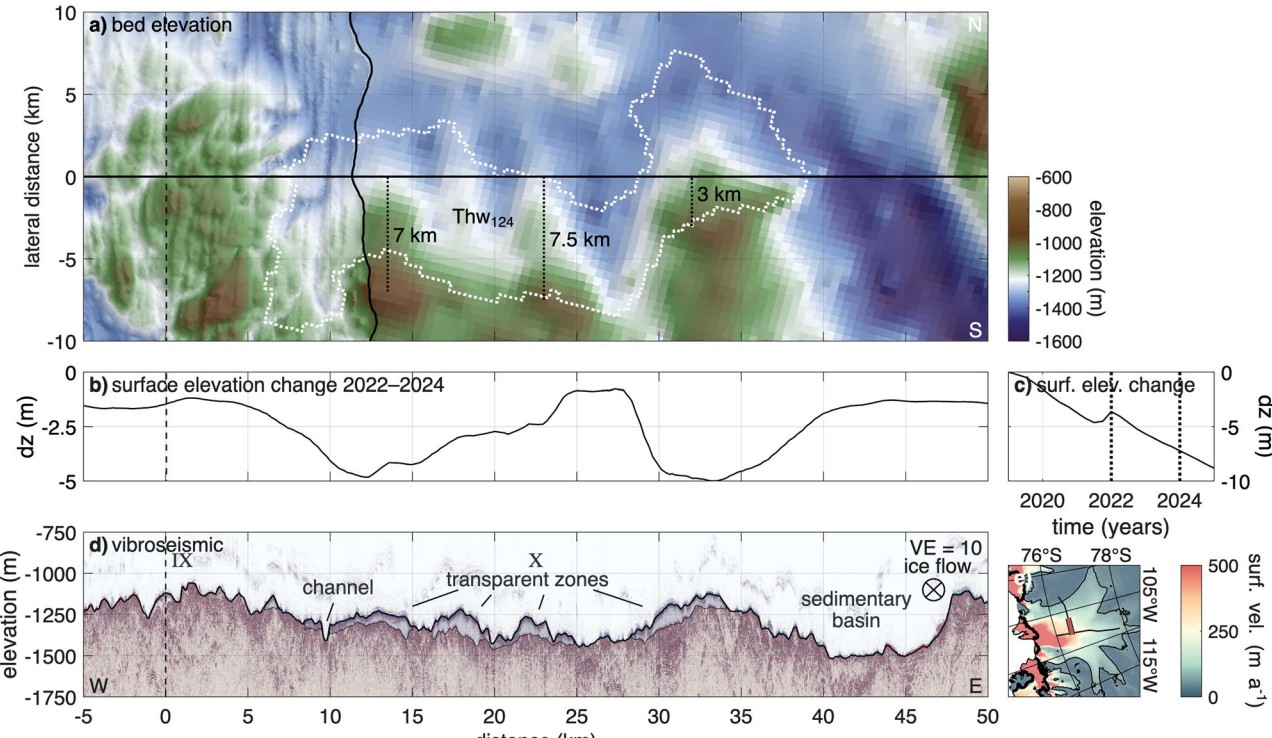

**Fig. 5 | Region IX and X of the across-flow profile. a** Map view of bed elevation (WGS 84) from Bedmap3[43,44] and swath airborne radar (black outline[35]) in lateral direction. Ice flow is from bottom to top. The white dotted line marks the outline of the subglacial lake Thw$_{124}$. The black dotted lines show the distances of the elevated topography to the vibroseismic profile. **b** Surface elevation change from 2022 to beginning of 2024 along the vibroseismic profile. **c** Time series of average surface elevation change of the lake area from 2019 to beginning of 2025. The two dotted lines mark the time period of the surface elevation change in **b**. **d** Kirchhoff migrated vibroseismic section with a vertical exaggeration VE = 10. Ice flow into the page. The vertical dotted lines visually connect the largest parts of the transparent zones with the lee sides of elevated topography in **a**. The shown profiles are a function of distance to the along-flow profile. Ice flow into the page. **e** Map of Thwaites Glacier with location of vibroseismic section (red line). Details are presented in Fig. 1c.

we observe hard, steep stoss sides and flat, softer lee sides—characteristic of crag-and-tail formations. The lower elevation and smoother areas exhibit a heterogeneous composition, including large basins of consolidated material as well as high porosity dilatant till.

Recently developed methods allow the inference of sliding laws directly from seismic-derived acoustic impedance[51]. Consequently, our dataset and observations have the potential to inform the sliding laws over an unprecedented area and in high detail, which will improve predictive models of Thwaites Glacier's dynamics and its contribution to sea-level rise. However, determining the sliding laws that are most appropriate for our soft bed/hard bed classification is beyond the scope of this study.

By applying a similar framework to our seismic observations, we aim to derive region-specific sliding laws that more accurately capture effective pressure and frictional behaviour, thereby enabling improved basal drag parametrisations in ice-sheet models. However, this goes beyond the scope of this study.

Our findings are consistent with previous seismic observations at LTR and UTR, which revealed a widespread pattern of hard beds on stoss sides and soft beds on lee sides[32–34]. These studies found crag-and-tails in the elevated regions and sedimentary basins with water pockets in deeper regions. The location of the sedimentary basins upstream and downstream of LTR is in agreement with those identified by aerogeophysical observations[30].

## Crag-and-tails

Crag-and-tail features are widespread beneath ice sheets, as shown by both seismic and airborne radar observations. Their formation is driven by ice flow over obstacles, generating high-pressure zones on the stoss side that expel subglacial water and transport sediment downstream to low-pressure

zones on the lee side. This process results in erosion upglacier of the obstacle (forming moats) and sediment deposition on the lee side, forming stratified tails[52]. Our impedance analysis indicates that the stoss sides are hard and likely composed of bedrock. Reflection coefficients are consistently higher on the stoss than on the lee side, although slope effects may influence this comparison. The lee-side tails exhibit high interval velocities, suggesting consolidated material overlain by dilatant till. We interpret this as progressive sediment deposition and compaction, where sediments are eroded from the stoss side, transported to the lee side, and gradually consolidated beneath newer, softer layers. Farther downstream, draping reflections indicate a diminishing sediment supply.

Using high-resolution airborne radar, crag-and-tails upstream of UTR, extending up to 5 km in width and reaching heights of 200–300 m, have been mapped[35]. The along-flow vibroseismic profile reveals distinct crag-and-tails just upstream of the ridges (km 137 to 143, Supplementary Fig. 2, 3), where airborne radar-derived bed topography shows several parallel features aligned with ice flow. These tails span 3–4.5 km in length, are ~700 m wide, elevated tens of meters above the surrounding terrain, and spaced roughly 1 km apart. The nearby crag-and-tails generate off-nadir reflections in the vibroseismic profile beneath the primary bed return. Vibroseismic data also reveal isolated crag-and-tails at the highest ridge elevations. These are slightly shorter, have steeper tails, and lack accompanying parallel features, making them difficult to identify from topography alone.

Crag-and-tails have also been discovered offshore in deglaciated palaeo-ice beds[53]. Within 120 km north of Thwaites Glacier's grounding line, crag-and-tail characteristics with comparable dimensions to those identified inland have been observed[54]. Preliminary sediment core and sediment echosounder analyses from an offshore crag-and-tail in the

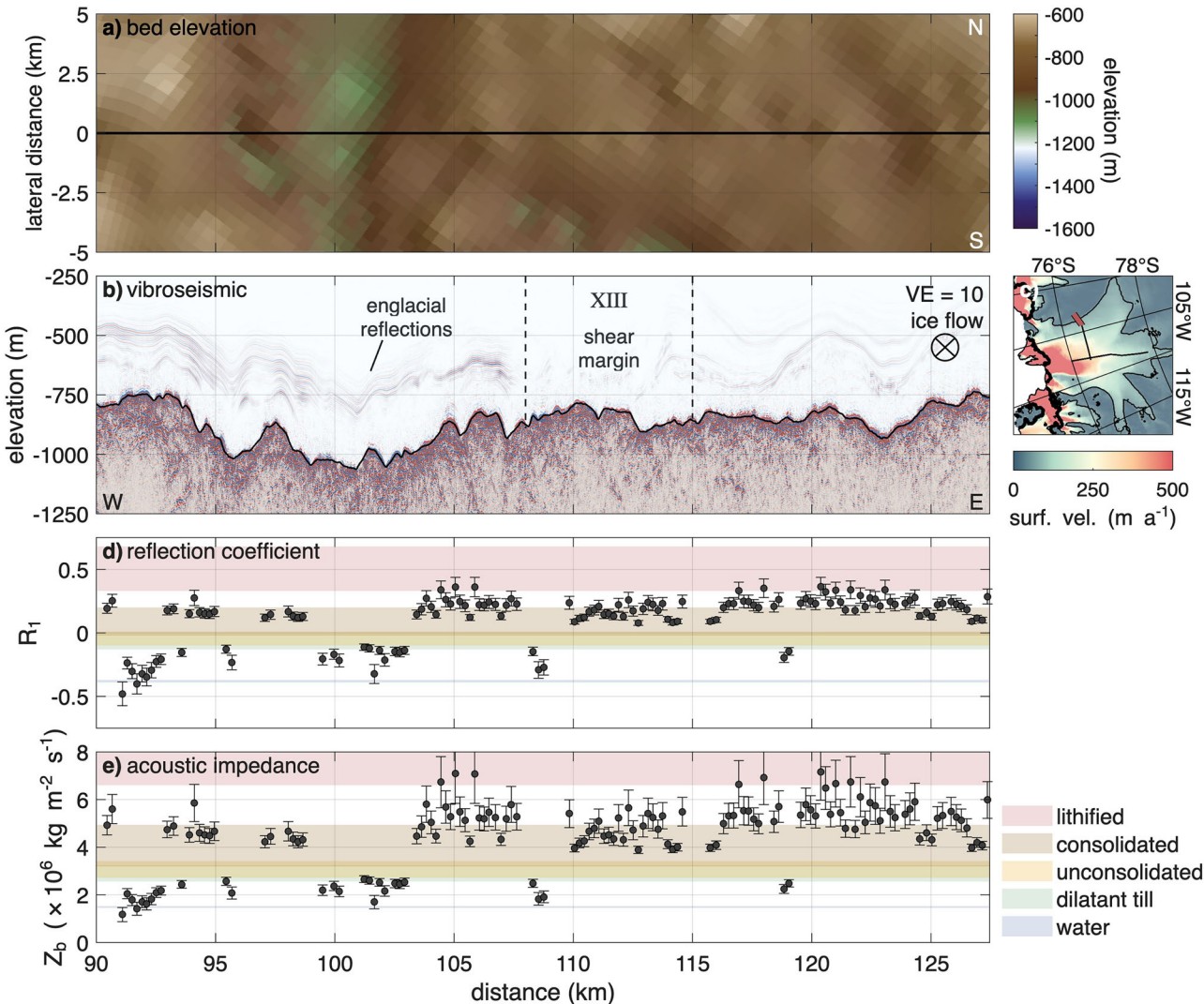

**Fig. 6 | Region XIII of the across-flow profile. a** Map view of bed elevation (WGS 84) from Bedmap3[43,44] in lateral direction **b** Kirchhoff migrated vibroseismic section with a vertical exaggeration VE = 10. The two vertical dashed lines mark the locations where the surface ice flow velocities are 50 (west) and 10 m a$^{-1}$ (east). **c** Map of Thwaites Glacier with location of vibroseismic section (red line). Details are presented in Fig. 1c. **d** Basal reflection coefficient $R_1$. **e** Basal acoustic impedance $Z_b$. The shown profiles are a function of distance to the along-flow profile. Ice flow into the page. Error bars indicate uncertainties in the reflection coefficient and acoustic impedance, as described in the Methods.

Bellingshausen Sea (RV Polarstern cruise PS134, 2022/23[55]) reveal consolidated layers beneath softer sediments in the tail (pers. comm. J. Klages, 2025), which is consistent with our seismic observations inland.

### Subglacial lake
Our vibroseismic observations provide the initial evidence suggesting that the subglacial lake Thw$_{124}$ is composed of highly porous, water-saturated sediments. The ranges of acoustic impedance and interval velocities limit our ability to further quantify porosity and water content within the lake sediments. The lake reaches up to 160 m in thickness and persists despite substantial surface lowering since 2019, indicating that drainage events only partially reduced the stored water volume. The seismic transparency and absence of internal reflections indicate a homogeneous, unstratified sediment-water mixture. Continuous water flow prevents the layering or consolidation. These deposits are located downstream of bedrock highs (Fig. 5), forming soft tails similar in geometry—but not composition—to crag-and-tail structures observed at the ridges. Comparable acoustically transparent features have also been observed by sonar surveys on the sea floor in front of Pine Island and Thwaites Glaciers; Evans et al. (2006)[56]

identified MSGLs consisting of thin, acoustically transparent sediments underlain by a strong basal reflector. Sediment cores from analogous features near the Antarctic Peninsula revealed that these transparent layers are composed of soft, highly porous (39% on average) till with massive diamicton, overlying stiff, low-porosity till[57]. The similarity in seismic character suggests that the subglacial lake consists of a comparable soft, water-rich substrate. The partial drainage since 2019 reduces the water content by ~10%, which could modestly increase basal drag and cause a slight deceleration in ice flow. This is consistent with minor ice-speed variations of <3% following subglacial lake filling or drainage[37,38]. The persistence of these soft tills, even after drainage events, may be comparable to terrestrial wetlands or aquifers, where water-saturated sediments remain despite fluctuations in water level.

### Shear margin
The characteristics of the bed beneath Thwaites Glacier play a key role in modulating future ice acceleration and mass loss. However, ice flux is not governed by velocity alone; it also depends on ice thickness and glacier width. Consequently, shear margin migration—driven by thinning in the

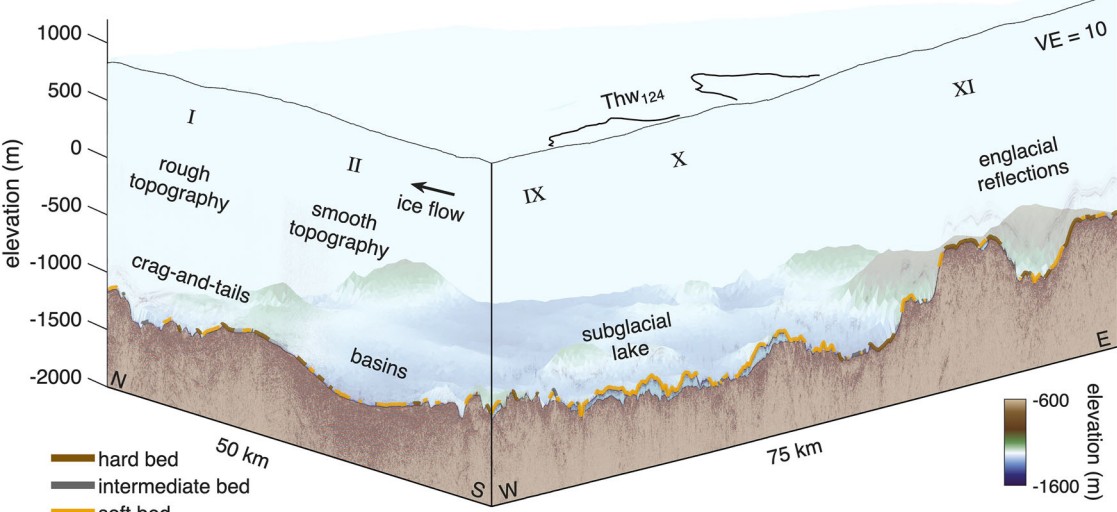

**Fig. 7 | Three-dimensional visualisation of vibroseismic profiles and derived basal properties.** View of the two crossing vibroseismic profiles together with the classification of basal conditions into hard, intermediate, and soft beds (colour). Key subglacial and englacial features identified from the seismic interpretation are annotated. The black lines at the glacier surface show the outline of Lake Thw$_{124}$ as derived from surface elevation changes. The background shows the bed elevation (WGS 84) from Bedmap3[43,44]. The image is vertically exaggerated by a factor of 10, and ice flow is from south to north.

main trunk—may contribute to future ice loss[58]. While the western shear margin is topographically constrained, the controlling factor on the eastern margin remains less clear[59]. Our across-flow profile shows that the eastern margin lies on an elevated plateau with no major elevation changes between fast- and slow-flowing side. Acoustic impedance values reveal a generally hard bed across the shear margin, composed of consolidated and lithified sediments, interspersed with isolated patches of soft bed. The bed character is similar in the regions immediately outside and inside the shear margin. These observations support previous airborne radar findings[59], which also showed no clear topographic or material control on the location of the eastern shear margin. Radar data indicated slightly elevated bed reflectivity and roughness on the slow-flowing side, insufficient to explain the shear margin's position[59]. Our seismic results reinforce the interpretation that neither basal topography nor substrate composition solely determines the margin's location.

### Ice crystal anisotropy

The viscosity of ice is influenced not only by the ice temperature but also by the mechanical anisotropy of the crystal orientation fabric[60,61]. Besides bed properties, the location of a shear margin can thus be influenced by ice rheology. Fabric anisotropy can be partially determined using radar and seismic measurements, as the propagation velocity of electromagnetic or elastic waves is dependent on the orientation of the measurement. The observed differences in the measured stacking velocities between the along-flow and across-flow profiles (Supplementary Figs. 12, 13) indicate a pronounced fabric anisotropy[62], particularly evident in the shear margin. Outside the shear margin, the stacking velocity decreases abruptly, suggesting a weak fabric anisotropy. These findings are consistent with fabric analysis from airborne radar data. The highest fabric strength was found in the centre of the shear margin, while low strength, indicating near-isotropic ice, was observed at the slow-flowing side[63]. At the fast-flowing side, the radar data reveal a stronger fabric.

Another observation that allows conclusions to be drawn regarding a change in crystal orientation fabric with depth are englacial reflections. Strong englacial reflections, located a few hundred metres above the ice base, comparable to those observed in both profiles, are widespread in West and East Antarctica and were linked to abrupt changes in the crystal orientation fabric[64,65]. However, the impact of crystal fabric anisotropy on the ice dynamics of the Thwaites Glacier and on its shear margin location remains an area of ongoing research. Further studies, including observations and

simulations, will be necessary for a better understanding of the ice rheology and its impact on ice flow dynamics on Thwaites Glacier.

### Conclusion

Our vibroseismic survey provides insights into the complex subglacial environment beneath Thwaites Glacier and its influence on ice flow dynamics. We identify four key findings:

1. *The bed beneath Thwaites Glacier is highly heterogeneous and critically controls ice flow dynamics.* Elevated, rough-topography regions (I, III, VI, VIII) alternate with areas of lower-elevation that consist of sedimentary basins (II, IV, and parts of V) and MSGLs (upstream III and VII). Ridges with the crag-and-tail features generate substantial basal drag and act as key resistive elements to fast ice flow. Such variations in basal roughness and bed properties across multiple scales strongly influence ice-flow acceleration, inland thinning, and grounding-line retreat. Crag-and-tail landforms are particularly prevalent and important structures that were also found offshore in palaeo-ice beds.

2. *Subglacial water bodies are widespread and morphologically diverse.* An active subglacial lake beneath Thwaites Glacier consists of highly porous, water-saturated sediments, reaching up to 160 m in thickness. Surface elevation changes capture only minor variations in lake volume, implying substantial water remains stored within the sediment matrix during drainage events. For the first time, we directly image the subglacial hydrology across several hundred kilometres, revealing that water-filled sediments occur predominantly in bedrock depressions, with smaller and more isolated patches also present along and on top of topographic highs.

3. *Current models do not adequately account for the complexity of the bed beneath Thwaites Glacier.* Water-bearing sediments and steep basal slopes are not adequately captured in current simulations, despite influencing basal drag and grounding line retreat. Our observations of basal mechanical properties and topography hold the potential to substantially enhance model constraints, thereby partly mitigating uncertainties associated with projections of future ice sheet mass loss.

4. *Further research is needed to understand the impact of the directional viscosity on ice flow dynamics of Thwaites Glacier.* Seismic wave speed variability across the glacier indicates a strong orientation of ice crystal fabric, particularly near the shear margins, which can affect the ice

deformation. Widespread englacial reflections above the bed may reflect variations in ice crystal fabric, but their analysis goes beyond the scope of this study.

Together, these findings highlight the need for large-scale, high-resolution seismic observations to reduce uncertainties in projections of ice sheet evolution. Although it will not be possible to obtain area-wide data coverage to directly infer basal conditions, statistical extrapolations hold the potential to infer such properties from observations[66]. Incorporating both detailed bed properties and ice-rheology into numerical models will be critical for improving projections of global sea-level rise from Thwaites Glacier and West Antarctica.

## Methods

### Data acquisition

As part of the International Thwaites Glacier Collaboration's project GHOST (Geophysical Habitat of Subglacial Thwaites) vibroseismic data were collected on Thwaites Glacier in summer 2022–23 and 2023–24. In the first season, we profiled a line parallel to the average flow direction (210 km long, hereafter *along-flow profile*) and in 2023–24 a line perpendicular to the flow direction (134 km long, hereafter *across-flow profile*). The seismic source was generated by a seismic vibrator (Industrial Vehicles Inc. EnviroVibe) placed on a polyethylene sled, all of which was towed by a snow tractor[67]. The seismic vibrator was operated at 60% peak force (40 kN) with a shot interval of 75 m. At every shot location, we performed two sweeps. The first sweep (setup sweep) was used for snow densification and not recorded, while the second sweep (acquisition sweep) was recorded with a 1500 m long snowstreamer towed behind the seismic vibrator[67]. For the along-flow profile, the sweep had a frequency range of 10–250 Hz, and for the across-flow profile, the frequency range was 10–200 Hz. The up-sweep length was 10 s, with 0.5 s tapering at begin and end. The snowstreamer consists of 60 channels with a 25 m group spacing. The eight gimballed p-wave geophones per channel recorded for 14 s at a sampling interval of 0.5 ms. With this set-up, we collected ten-fold data at up to 24 km per day. The along-flow profile was collected in the upstream direction and the across-flow profile from west to east. Details are presented in Supplementary Table 1.

### Vibroseismic processing

The processing of the recorded vibroseismic data to produce a migrated section involves several standard steps outlined below. The recorded raw vibroseismic data were compressed by a cross-correlation with a synthetic source signal. The individual traces were assigned geometry from shot locations and group intervals to organize common midpoint (CMP) gathers. Next, noisy or dead traces were removed. In particular, we removed the 8 channels closest to the source as these were generally clipped. A frequency filter from 10–190 Hz was applied to reduce high-frequency noise, and a notch filter at 190 Hz was applied to reduce ringing from the spurious response of the geophones. We performed a zero-phase spike deconvolution to compress the wavelet to a spike. Afterwards, we re-sorted the data into CMP gathers and determined the stacking velocities (Supplementary Methods 1.1, Supplementary Figs. 12–14). The resulting velocity field was used to perform the normal moveout (NMO) correction. We stacked the NMO-corrected CMP gathers to improve the signal-to-noise ratio. To remove diffraction hyperbolas, we migrated the stacked section with a time–space Kirchhoff migration using interval velocities of ice, geological features and the bed (Supplementary Methods 1.2, Supplementary Figs. 12, 13). Finally, we applied a two-way traveltime (TWT) to depth conversion with the interval velocities and obtained the elevation relative to WGS84 based on ice surface heights that we measured with a GNSS system at the source.

### Amplitude analysis

We used amplitude analysis to derive information regarding the geological characteristics of the materials at the ice base[68–71]. This analysis is conducted through the estimation of the reflection coefficient and the acoustic impedance. The reflection coefficient $R$ at an interface between two media quantifies the ratio of the amplitude of the incident to the reflected waves. The basal reflection coefficient $R_1$ for small offsets was calculated for each CMP as follows:

$$R_1 = \frac{A_1}{A_0}\frac{1}{\gamma}e^{\alpha d_1}, \; \gamma = \frac{1}{d_1} \tag{1}$$

[72], with the basal amplitude $A_1$, the source amplitude $A_0$, the spreading loss $\gamma$, the two way travel distance $d_1$ of the basal return and the attenuation constant $\alpha = (2.6 \pm 0.5) \times 10^{-4}\,\mathrm{m}^{-1}$ (Supplementary Methods 2, Supplementary Figs. 15–18).

We eliminated outliers of $R_1$ by filtering for polarities. To determine the dominant polarity, we averaged polarities using a moving window of 16 CMPs (200 m along the profile), approximately matching the Fresnel zone width (~220 m). If 80% or more of the CMPs had the same polarity, we removed the conflicting ones and averaged the remaining amplitudes and reflection coefficients within that group (200 m window). Next, we manually eliminated groups with ambiguous or inconsistent basal returns, and those intersected by diffraction hyperbola that interfere with the reflection amplitude. The error of $R_1$ was then calculated as

$$\delta R_1 = \frac{A_1 e^{\alpha d_1}}{A_0 \gamma_1}\sqrt{\left(\frac{\delta A_0}{A_0}\right)^2 + \left(d_1 \delta \alpha\right)^2} \tag{2}$$

with the uncertainties of the source amplitude $\delta A_0$, and of the attenuation $\delta \alpha$.

The reflection coefficient $R$ is an estimate of the contrast in acoustic impedance ($Z = \rho V_p$) between the media interfaces:

$$R = \frac{Z_2 - Z_1}{Z_2 + Z_1} = \frac{\rho_2 V_{p2} - \rho_1 V_{p1}}{\rho_2 V_{p2} + \rho_1 V_{p1}} \tag{3}$$

where $\rho$ is the density, and $V_i$ the p-wave velocity in the upper ($i = 1$) and lower ($i = 2$) layers. Assuming an acoustic impedance of the basal ice of $Z_{\mathrm{ice}} = 3.33 \pm 0.04 \times 10^6\,\mathrm{kg\,m^{-2}\,s^{-1}}$ [73], we can calculate the acoustic impedance of the basal material $Z_b$ based on the reflection coefficient:

$$Z_b = Z_{\mathrm{ice}}\frac{1 + R_1}{1 - R_1}. \tag{4}$$

The uncertainty of the acoustic impedance $\delta Z_b$ was derived from error propagation including the uncertainty of the reflection coefficient $\delta R_1$ and of the acoustic impedance of ice $\delta Z_{\mathrm{ice}}$:

$$\delta Z_b = \sqrt{\left(\frac{2 Z_{\mathrm{ice}} \delta R_1}{(1 - R_1)^2}\right)^2 + \left(\frac{(1 + R_1)\delta Z_{\mathrm{ice}}}{1 - R_1}\right)^2}. \tag{5}$$

Beyond the uncertainties explicitly accounted for in the impedance analysis, additional processes may influence the derived acoustic impedance, including basal slope effects, off-nadir reflections, and debris entrained within the basal ice. Based on the generally low bed slopes and the characteristics of the observed basal reflections, we expect these effects to be of secondary importance and unlikely to significantly bias the impedance-based bed classification.

The likely range of p-wave velocities, densities, reflection coefficients and acoustic impedance for different lithologies and hydrological settings is presented in Supplementary Table 2. In addition, we classify basal conditions into three categories based on inferred resistance to deformation, using acoustic impedance as a proxy for basal mechanical behaviour (Supplementary Table 3). We distinguish between a "hard bed", where basal motion is dominated by sliding at the ice-bed interface, and a "soft bed", where motion is primarily due to deformation of subglacial till. A "hard bed"

consists of consolidated and lithified sediments with an acoustic impedance $Z_b > 3.24 \times 10^6 \, \text{kg} \, \text{m}^{-2} \, \text{s}^{-1}$. In contrast, a "soft bed" consists of dilatant till with a lower impedance than unconsolidated, stiff sediments ($Z_b < 2.72 \times 10^6 \, \text{kg} \, \text{m}^{-2} \, \text{s}^{-1}$). We define the transition between soft and hard beds as an "intermediate bed", with acoustic impedance of $2.72 \times 10^6 < Z_b < 3.24 \times 10^6 \, \text{kg} \, \text{m}^{-2} \, \text{s}^{-1}$. For this class, the limited sensitivity of small-offset seismic acquisitions precludes a unique determination of the dominant basal deformation mechanism.

### Satellite-derived surface elevation changes

We generate an elevation time series from 2019 through austral winter 2025, and use this time series to isolate lake extent over the period of observations (Fig. 5b). Elevation models were derived by fitting surfaces of elevation change to ICESat-2 elevation retrievals relative to a reference elevation model from the first quarter of 2022[37]. The fitting procedure minimized an objective functional that considered data misfit, spatial gradients in the constructed reference elevation model, elevation-change rate fields, temporal gradients in elevation-change rate and the magnitude of model bias parameters[37,38]. We estimated the outline of the lake from the contour of surface elevation change between 2022 and 2024 that exceed a subsidence of 2.5 m.

### Data availability

The raw, stacked and migrated data of the vibroseismic measurements, as well as derived products of reflection coefficients and acoustic impedance, are published at the World Data Center PANGAEA (https://doi.org/10.1594/PANGAEA.987704)[74].

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

## Acknowledgements

This work is from the Geophysical Habitat of Subglacial Thwaites (GHOST) project, a component of the International Thwaites Glacier Collaboration (ITGC). We thank the members of the GHOST Team for their support during the field expeditions and subsequent discussions; a full list of members is provided in the Supplementary Information. We thank Florian Koch, Hannes Laubach, Catrin Thomas, Taff Raymond, Sasha Doyle, David Jamison, Peter Young, and Richard Pryce for their great support in the field. We thank Johann Klages for an insightful discussion on the geology of the crag-and-tails. The authors would like to thank Aspen Technology, Inc. for providing software licenses and support. Field work was supported through grants NSF PLR 1738934 and NERC NE/S006672/1, NE/S006621/1, NE/S006613/1 and AWI logistic grant AWI_ANT_31. OZ was funded by AWI's INSPIRES I program. We acknowledge support by the Open Access publication fund of Alfred Wegener Institute Helmholtz Centre for Polar and Marine Research.

## Author contributions

Ole Zeising performed the vibroseismic measurements, processed and analysed the vibroseismic measurements and wrote the manuscript. Olaf Eisen performed the vibroseismic measurements and contributed to data interpretation and discussion. Coen Hofstede performed the vibroseismic measurements and contributed to data interpretation and discussion. Ronan Agnew supported the processing and analyses of the vibroseismic measurements and contributed to data interpretation and discussion. Alex Brisbourne coordinated the GHOST project as well as the implementation of both field seasons and contributed to data interpretation and discussion. Andrew Hoffman analysed and interpreted the satellite-data to derive surface elevation changes. Sridhar Anandakrishnan coordinated the GHOST project as well as the implementation of both field seasons and contributed to data interpretation and discussion.

## Funding

## Competing interests

The authors declare no competing interests.
