## [Transparent Peer Review file · Communications Earth & Environment]

Hard rocks and deep wetlands beneath Thwaites Glacier in Antarctica

Corresponding Author: Dr Ole Zeising

Version 0:

Decision Letter:

Dear Dr. Zeising,

Your manuscript titled "Hard rocks and deep wetlands: characterizing the bed of Thwaites Glacier, Antarctica, from vibroseismic measurements" has now been seen by 3 reviewers, and we include their comments at the end of this message. They find your work of interest, but some important points are raised. We are interested in the possibility of publishing your study in Communications Earth & Environment, but would like to consider your responses to these concerns and assess a revised manuscript before we make a final decision on publication.

We therefore invite you to revise and resubmit your manuscript, along with a point-by-point response that takes into account the points raised. Please highlight all changes in the manuscript text file.

Please submit your point-by-point responses as a separate file, distinct from your cover letter where you can add responses to the Editors' comments that you do not want to be made available to the reviewers. Word files are preferred. We recommend that any figures, tables or graphs that are included in the response to reviewers are also included in the main article or Supplementary Information.

Please use the following link to submit your revised manuscript, point-by-point response to the referees' comments (which should be in a separate document to any cover letter), a tracked-changes version of the manuscript (as a PDF file) and the completed checklist:

Link Redacted

We hope to receive your revised paper within six weeks; please let us know if you aren't able to submit it within this time so that we can discuss how best to proceed. If we don't hear from you, and the revision process takes significantly longer, we may close your file. In this event, we will still be happy to reconsider your paper at a later date, as long as nothing similar has been accepted for publication at Communications Earth & Environment or published elsewhere in the meantime.

Please do not hesitate to contact us if you have any questions or would like to discuss these revisions further. We look forward to seeing the revised manuscript and thank you for the opportunity to review your work.

Best regards,

Zhendong Zhang, PhD
Editorial Board Member
Communications Earth & Environment
orcid.org/0000-0003-4689-1577

Nicola Colombo, PhD
Associate Editor, Communications Earth & Environment
Consulting Editor, Communications Sustainability

EDITORIAL POLICIES AND FORMATTING

- Behavioural and social science
- Ecological, evolutionary & environmental sciences
- Life sciences

Furthermore, please align your manuscript with our format requirements, which are summarized on the following checklist: <https://www.nature.com/documents/commsj-phys-style-formatting-checklist-article.pdf> Communications Earth & Environment formatting checklist

and also in our style and formatting guide <https://www.nature.com/documents/commsj-phys-style-formatting-guide-accept.pdf> Communications Earth & Environment formatting guide .

*** DATA: Communications Earth & Environment endorses the principles of the Enabling FAIR data project (<http://www.copdess.org/enabling-fair-data-project/>). We ask authors to make the data that support their conclusions available in permanent, publically accessible data repositories. (Please contact the editor if you are unable to make your data available).

All Communications Earth & Environment manuscripts must include a section titled "Data Availability" at the end of the Methods section or main text (if no Methods). More information on this policy, is available at <http://www.nature.com/authors/policies/data/data-availability-statements-data-citations.pdf>.

If a community resource is unavailable, data can be submitted to generalist repositories such as <https://figshare.com/> or <http://datadryad.org/> Dryad Digital Repository. Please provide a unique identifier for the data (for example a DOI or a permanent URL) in the data availability statement, if possible. If the repository does not provide identifiers, we encourage authors to supply the search terms that will return the data. For data that have been obtained from publically available sources, please provide a URL and the specific data product name in the data availability statement. Data with a DOI should be further cited in the methods reference section.

REVIEWER COMMENTS:

Reviewer #1 (Remarks to the Author):

This paper presents new information about the bed properties of Thwaites Glacier derived from seismic reflection observations on major along and across flow traverses. It shows for the first time the nature of the subglacial 'lake' noted from satellite observations, as well as constraining the distribution of subglacial water, un-consolidated till, consolidated till, and lithified bedrock. Imaging of the subglacial lake is highly significant, as the nature of dynamic subglacial lakes has been much debated, with traditional radar techniques often being unable to image them. The study has wider implications for models of the Thwaites Glacier system as it better constrains the basal properties – such observations can both directly

inform models of retreat and be used to guide inference of basal properties from more extensive airborne radar datasets.

I think this paper and the associated data will be highly significant for the work on ice sheet behaviour in the Thwaites Glacier region and beyond.

I have a few comments on the paper in terms of readability and suggestions of areas where the authors could consider going further with their interpretation set out in detail below.

L20-22 "Presently, Thwaites and Pine Island Glaciers have collectively an ice discharge into the ocean of $250 \pm 10\text{Gt a}^{-1}$, making them the largest regional contributors to global sea-level rise [13]". Strictly speaking the discharge is irrelevant to sea level rise – it is the imbalance in discharge vs accumulation (or increase in velocity with fixed accumulation) which drives sea level rise. Could be re-worded slightly to reflect this.

L98-99 "To the west, the bed rises steeply to +100m at the shear margin" This part of the sentence appears repeated over the page.

L127-128 Regarding "moats" upstream and especially downstream of tails. In Fig 3 at 124 km and 137 (for example) narrow v shaped notches are apparent. In plan view these seem to align with apparent erosive, probably water cut, channels running oblique to ice flow. These features therefore seem to reflect intersections between channels cut by sub-glacial rivers and sediment moved by the glacier. Rather than true "moats" which you would expect to surround a single feature. On the upstream side moats seems a more appropriate description, but down-stream this seems potentially inaccurate.

L132 – around this point it would be nice to see a zoom on a crag-tail feature either as an inset in Fig 3, or in the supplementary material.

179-180 "A 100m deep and 500m wide moat separates two of these zones in the western part near the LTR". Also see point around L127-128. Is this a moat or a continuation of a channel flowing off the adjacent highland? An associated interesting question which may be beyond the scope of this study - how is a 'moat/channel' cut into a lake? E.g. does sediment erosion create band of 'clean' transparent ice?

313-315 "These water bodies reach up to 160m in thickness and persist despite substantial surface lowering since 2019, indicating that drainage events only partially reduce the stored water volume." Any thought from either observations or literature on the porosity/percent water? How much do these marshes dry out for example in response to 10 m surface lowering. Could be important as for example 80% porosity to 70% porosity is still effectively soup (0 strength), but 15% to 5% could reflect a fundamental shift in strength of the basal material.

L369-370 "areas of lower-elevation that either consist of sedimentary basins (II, IV, and parts of V) or MSGLs (upstream III and VII)". Are sedimentary basins and MSGL mutually exclusive? can you not get MSGL in sedimentary basins and/or get MSGL which mask sedimentary basins? Do you get sedimentary basins with no MSGL? If this is correct it could be expanded further. My understanding from offshore data was that MSGL were pretty ubiquitous wherever ice flows rapidly over sediment.

Reviewer #2 (Remarks to the Author):

Review of Zeising et al.

Zeising et al. present detailed analysis of 344 km of vibroseismic profiles on Thwaites Glacier in West Antarctica. The dataset collected characterises basal properties and subglacial hydrology across an impressively-large and scientifically-important study area. The results will be key to guiding and refining future modelling efforts that aim to forecast dynamic mass loss from Thwaites. The analysis of crag and tail landforms, which play an important role in resisting ice flow, is a key aspect of this paper which could be emphasised further, perhaps with an additional annotated figure. The methods appear robust, the figures are of a high quality and support the arguments and conclusions made, and the writing is clear. This paper changes our understanding of the nature of the basal environment of Thwaites by showing how basal properties vary over a scale of 100s of kilometers both longitudinally and transverse to ice flow, and importantly across the shear margin of Thwaites. These properties are key to understanding glacier dynamics. Figure referencing in the main text could be increased slightly to help the reader navigate the paper, see specific comments below. I have only one major comment relating to the use of the term 'rheology' to describe hard beds, and a number of minor comments, both detailed below.

Samuel Doyle

Major Comment

Throughout the manuscript the term rheology is used to describe bed properties in a way that includes both hard bed sliding and deformation of subglacial sediments. It's not correct to refer to the rheology of hard bed sliding as hard beds do not flow. While it's correct to use the term for subglacial sediment deformation this usage also neglects ice-sediment decoupling

which can dominate basal motion at least in some locations (e.g. Iverson et al., 1995). I think all is required here is to use a different term to describe the nature of the bed in relation to basal motion.

Iverson, N. R., Hanson, B., Hooke, R. L., & Jansson, P. (1995). Flow mechanism of glaciers on soft beds. *Science*, 267(5194), 80-81.

Minor Comments

L20 – here ice discharge per annum is given for Thwaites Glacier and Pine Island Glacier combined and then it's stated that they are the largest regional contributors to global sea level rise. This raises two questions: why combine the discharge estimate for Thwaites with another glacier that wasn't studied? Second, why state that they are the largest contributors in this region? Few readers will understand the boundaries of this (possibly undefined) region in a way that allows them to make sense of the importance of Thwaite's annual ice discharge globally. It would be better to state the global sea level contribution of Thwaites on its own and then give it as a percentage of total global ice mass loss.

L89 – Is GR defined earlier?

L89 – I presume all elevations are relative to the WGS-84 ellipsoid as defined later. Why weren't orthometric heights used? Is there a major difference between the geoid and WGS-84 ellipsoid in the study area?

L100 – 'abruptly over some 10 km' – over 10 km doesn't seem very abrupt but perhaps this is valid over the length scale of the seismic line. In any case, avoid the informal 'some' as this would usually be used to refer to some 10s of km, which isn't abrupt.

Figure 1 – capitalise 'Surf. Vel. on (a).

L113 – UTR has previously been defined so doesn't need to be defined again here.

L113 – Add "at" to "located at km 115-145"

L128 – The moat on Figure 5 at km 10 looks deeper than 12 m. It would be good to expand on moats and crag and tails here or in the introduction so the reader understands what they are prior to their mention in the discussion. Moats aren't a well known glacial landform as far as I'm aware so a brief introduction is warranted.

L138 – Give figure reference for stratification.

Figure 2 caption – expand on what this unpublished dataset is in the manuscript.

L169 – state which seismic line with an appropriate label (e.g. E-W or transverse).

L160 – Give figure reference to Figures 5 and S5. Same applies to similar sections below.

L170 – can the subglacial lake locations be marked on Figure S5?

L169 "covering" -> "cover"

L185 – add "elevation" to "bed elevation".

L217 – where specifically are you referring to outside of Thwaites Glacier – give km reference.

L224 – Bow tie structures are mentioned but never explained or expanded on. Can this be addressed briefly?

L231 – the last sentence in this paragraph isn't a complete sentence and needs revising and the point made needs expanding on to be worthwhile including.

L238 – The sentence starting 'seismic measurements are essential' can be omitted as this argument was established in the introduction.

L243 – Go further in this first sentence and state that the topography and roughness constrained by the results presented improves on previous datasets – tell the reader here what you go on to tell them later in the paragraph.

L243 – Consider whether the phrase 'the bed is characterised by significant roughness and large bed slopes' is excessively verbose in comparison to something like "the bed is rough and steeply sloping."

L247 – qualify the new sentence with "In these simulations,"

L258 – Remove repeated last sentence.

L274 and L277 – omit 'also'

L289 – This section on crag and tails is excellent and a key aspect of the paper. Can you give a figure reference to an example in the seismogram? Or maybe even create a new figure of a crag and tail example with more detail and annotations.

L298 – Figure reference and km reference here would help the reader.

L304 – New paragraph here.

L314 – plural ‘water bodies’ – singular lake. Choose one of these?

L315 – use past tense ‘reduce’

L320 – lower case “sonar”

L321 – The sentence starting Evans et al. appears to directly follow on from the previous. If so, link these explicitly with a semi-colon.

L336 – add “shear” to “shear margin”.

L374 – Make “structure” plural and change “was” to “were”.

L379 – this sentence says you observed water in both depressions and hills but it would be useful to know whether the former was more common.

L395 – citation for statistical extrapolations.

L419 – ‘do’ to ‘to’

L462-466 – correct reverse quotation marks. Remove ‘as’ on L465.

L471 – give a figure reference for the surface elevation data in this section.

L649 – give link to data repository.

Supplementary Material

L10 – mark new paragraph with either a vertical space between paragraphs or an indent.

L37 – what geological features are being referred to?

Reviewer #3 (Remarks to the Author):

This manuscript presents an extensive and technically sophisticated vibroseismic investigation of the Thwaites Glacier bed, offering new insights into basal topography, rheology, and hydrology. The study’s scope—344 km of seismic profiling—is impressive and contributes valuable empirical data to one of the most critical regions for ice-sheet stability and sea-level projections. The integration of amplitude, velocity, and impedance analyses, combined with cross-referencing to radar and altimetry datasets, demonstrates strong methodological rigor. It is also great to see that the data are uploaded to World Data Center PANGAEA.

However, while the work is clearly of high potential significance, several areas could be strengthened to enhance clarity, reproducibility, and scientific impact:

1. The introduction is well contextualized within the literature on Thwaites Glacier, but the novelty of this study is not fully articulated. The paper often reiterates that current models lack the complexity revealed by the new seismic data, yet it could more explicitly state what specific knowledge gaps this study closes compared to prior seismic work (e.g., Muto et al. 2019; Clyne et al. 2020). A concise statement of unique contributions—whether in spatial coverage, resolution, or methodological innovation—would strengthen the manuscript’s conceptual foundation.
2. Although the amplitude and impedance analyses are detailed, the manuscript provides limited discussion of uncertainty quantification and its influence on geological interpretation. Given the heterogeneous basal conditions, explicit error ranges for impedance-derived classifications (“hard,” “soft,” “water-saturated”) are essential. For instance, slope effects, off-nadir reflections, and local anisotropy could bias impedance contrasts; these sources of error should be more systematically evaluated. Moreover, some interpretations (e.g., identifying “water-saturated sediments” vs. “unconsolidated till”) rely heavily on qualitative comparison to prior work and could benefit from more quantitative justification.
3. The discussion emphasizes that current models underrepresent basal complexity but stops short of connecting these findings to specific modelling frameworks or parameters. How might the new impedance-derived rheology maps alter basal drag formulations or grounding-line stability simulations? A schematic or conceptual model linking seismic observations to potential model parameterization would make the results more actionable for ice-sheet modelers.
4. The numerous figures are technically rich but sometimes dense. Readers may struggle to extract key spatial relationships without frequent cross-referencing between figures 1–6. Simplified summary figures or conceptual diagrams synthesizing major bed types (hard rock, sediment basins, water pockets) along both profiles would improve readability.

5. The section on englacial reflections introduces an intriguing discussion of ice fabric anisotropy but remains speculative. The link between seismic reflections, fabric orientation, and shear-margin rheology should either be supported with more direct data or framed more cautiously. Distinguishing between englacial scattering, layer reflectivity, and genuine crystal orientation effects requires stronger methodological justification. In particular, it will be helpful to have a representative waveform figure to clearly show the englacial reflections along with other key phases, such as in Fig.S12.

** Visit Nature Portfolio's author and referees' website at www.nature.com/authors for information about policies, services and author benefits**

Communications Earth & Environment is committed to improving transparency in authorship. As part of our efforts in this direction, we are now requesting that all authors identified as 'corresponding author' create and link their Open Researcher and Contributor Identifier (ORCID) with their account on the Manuscript Tracking System prior to acceptance. ORCID helps the scientific community achieve unambiguous attribution of all scholarly contributions. You can create and link your ORCID from the home page of the Manuscript Tracking System by clicking on 'Modify my Springer Nature account' and following the instructions in the link below. Please also inform all co-authors that they can add their ORCIDs to their accounts and that they must do so prior to acceptance.

Version 1:

Decision Letter:

Dear Dr Zeising,

Your manuscript titled "Hard rocks and deep wetlands: characterizing the bed of Thwaites Glacier, Antarctica, from vibroseismic measurements" has now been seen by our reviewers, whose comments appear below. In light of their advice we are delighted to say that we are happy, in principle, to publish a suitably revised version in Communications Earth & Environment.

We therefore invite you to revise your paper one last time to address the remaining concerns of our reviewers. At the same time we ask that you edit your manuscript to comply with our format requirements and to maximise the accessibility and therefore the impact of your work.

EDITORIAL REQUESTS:

*****Please take care to match our formatting and policy requirements. We will check revised manuscript and return manuscripts that do not comply. Such requests will lead to delays. *****

SUBMISSION INFORMATION:

OPEN ACCESS:

Communications Earth & Environment is a fully open access journal. Articles are made freely accessible on publication. For further information about article processing charges, open access funding, and advice and support from Nature Portfolio,

please visit <https://www.nature.com/commsenv/open-access>

Link Redacted

Best regards,

Zhendong Zhang, PhD
Editorial Board Member
Communications Earth & Environment
orcid.org/0000-0003-4689-1577

Nicola Colombo, PhD
Associate Editor, Communications Earth & Environment
Consulting Editor, Communications Sustainability

REVIEWERS' COMMENTS:

Reviewer #1 (Remarks to the Author):

Second review of "Hard rocks and deep wetlands: characterizing the bed of Thwaites Glacier, Antarctica, from vibroseismic measurements".

This paper presents new seismic reflection data from the central part of Thwaites Glacier. The authors have responded well to my previous comments and I am happy to see the paper move forward to publication.

One minor note – Line 119 "LTR has the highest elevation of 1050m, surpassing its surroundings by approximately 350 to 400m". I think this should be an elevation of -1050m.

Reviewer #2 (Remarks to the Author):

My previous comments have been satisfactorily addressed. I have no further comments.

Reviewer #3 (Remarks to the Author):

The revised manuscript thoroughly address my comments, I appreciate the new Fig.7. I have no further concern to the manuscript and ready to sign off. Congradulations to the authors for a nice piece of work!

** Visit Nature Portfolio's author and referees' website at <http://www.nature.com/authors> for information about policies, services and author benefits**

Hard rocks and deep wetlands: characterizing the bed of Thwaites Glacier, Antarctica, from vibroseismic measurements

Point-by-point response

Dear Reviewers,

We would like to thank you for your efforts to improve the quality and clarity of our manuscript.

We have carefully considered all suggestions and revised the manuscript accordingly. In the revised version, we have:

- Improved the description of the geological features and clarified the criteria used for bed classification.
- Incorporated the term “dilatatant till” consistently throughout the manuscript text and figures.
- Added new figures to both the main manuscript and the Supplementary Information: (1) a detailed zoom of a crag-and-tail feature (Supplementary Figure 3), (2) a simplified summary figure synthesizing the main results (Figure 7), and (3) a bed classification for both seismic profiles (Supplementary Figure 11).
- Added a new Supplementary Video for illustration of the seismic profiles in the context of Thwaites Glacier.
- Added a new paragraph outlining how the basal drag can be derived from the acoustic impedance values presented in this study.
- Improved overall readability and more clearly highlighted the novelty and significance of the work.
- Corrected typos and minor inconsistencies throughout the manuscript.

We believe that these revisions have substantially strengthened the manuscript and addressed the reviewers’ concerns. We appreciate your insightful feedback and hope that the revised version meets your expectations.

Thank you once again for your time and expertise.

Kind regards,
Ole Zeising and co-authors

Reviewer #1 (Remarks to the Author):

This paper presents new information about the bed properties of Thwaites Glacier derived from seismic reflection observations on major along and across flow traverses. It shows for the first time the nature of the subglacial ‘lake’ noted from satellite observations, as well as constraining the distribution of subglacial water, un-consolidated till, consolidated till, and lithified bedrock. Imaging of the subglacial lake is highly significant, as the nature of dynamic subglacial lakes has been much debated, with traditional radar techniques often being unable to image them. The study has wider implications for models of the Thwaites Glacier system as it better constrains the basal properties – such observations can both directly inform models of retreat and be used to guide inference of basal properties from more extensive airborne radar datasets.

I think this paper and the associated data will be highly significant for the work on ice sheet behaviour in the Thwaites Glacier region and beyond.

I have a few comments on the paper in terms of readability and suggestions of areas where the authors could consider going further with their interpretation set out in detail below.

L20-22 “Presently, Thwaites and Pine Island Glaciers have collectively an ice discharge into the ocean of $250 \pm 10\text{Gt a}^{-1}$, making them the largest regional contributors to global sea-level rise [13]”. Strictly speaking the discharge is irrelevant to sea level rise – it is the imbalance in discharge vs accumulation (or increase in velocity with fixed accumulation) which drives sea level rise. Could be re-worded slightly to reflect this.

Response:

Thanks for your comment. We agree and changed the sentence in the revised version to: *L20–22: “Since 2002, the Thwaites drainage basin has lost mass at an average rate of $60.5 \pm 9.2\text{ Gt/a}$, corresponding to a global mean sea-level rise of $0.17 \pm 0.03\text{ mm/a}$ (or 5%), making it the largest Antarctic contributor to sea-level rise (Groh & Horwath, 2021, Horwath et al., 2025).”*

Groh, A., & Horwath, M. (2021). Antarctic Ice Mass Change Products from GRACE/GRACE-FO Using Tailored Sensitivity Kernels. *Remote Sens.*, 13(9), 1736. doi:10.3390/rs13091736

Horwath, M., Döhne, T. & Knöfel, C. Gravimetric Mass Change, Antarctic Ice Sheet Project, ESA Climate Change Initiative [data set] (2025). URL https://data1.geo.tu-dresden.de/ais_gmb/. Last access: 05 December 2025.

L98-99 “To the west, the bed rises steeply to +100m at the shear margin” This part of the sentence appears repeated over the page.

Response:

Thanks for noticing! We removed the first part.

L127-128 Regarding “moats” upstream and especially downstream of tails. In Fig 3 at 124 km and 137 (for example) narrow v shaped notches are apparent. In plan view these seem to align with apparent erosive, probably water cut, channels running oblique to ice flow. These features therefore seem to reflect intersections between channels cut by sub-glacial rivers and sediment moved by the glacier. Rather than true “moats” which you would expect to surround a single feature. On the upstream side moats seems a more appropriate description, but down-stream this seems potentially inaccurate.

Response:

Thanks for your comment. We agree that only the upstream feature of a crag is a “moat”, while the similar looking graben-like feature behind the tail should not be called “moat” due to a different formation mechanism. In addition, we corrected a mistake in this sentence: The moat is 120 m deep, not 12 m.

In the revised version, we have changed the sentence to:

L130–132: “We observed moats up to ~120 m deep upstream of crags (Alley et al., 2023, Schlegel et al., 2024) and similar features but likely with a different formation mechanism downstream of tails.”

Alley, R. B. et al. Ghostly flute music: drumlins, moats and the bed of thwaites glacier.597 *Annals of Glaciology* 63, 153–157 (2023).

Schlegel, R. et al. Subglacial bedform and moat initiation beneath Rutford Ice Stream, West Antarctica. *Geomorphology* 458, 109207 (2024).

L132 – around this point it would be nice to see a zoom on a crag-tail feature either as an inset in Fig 3, or in the supplementary material.

Response:

Thanks for your suggestion. We have added a zoom on a crag-and-tail feature in the Supplementary Information as a new Supplementary Figure 3.

Supplementary Figure 3: Crag-and-tail in region III of the along-flow profile. Figure panels and variables are identical to those described in Supplementary Figure 1.

L179-180 “A 100m deep and 500m wide moat separates two of these zones in the western part near the LTR”. Also see point around L127-128. Is this a moat or a continuation of a channel flowing off the adjacent highland? An associated interesting question which may be beyond the scope of this study - how is a 'moat/channel' cut into a lake? E.g. does sediment erosion create band of 'clean' transparent ice?

Response:

Thanks for your comment. We agree that this feature is rather a “channel” than a “moat”. We have updated the text and the figure.

How a channel can exist in a lake is indeed a very interesting question. To find an answer for this particular case, a 3D seismic image would be needed to reveal the topography of the water-bed. However, this goes beyond the possibilities of this study.

313-315 “These water bodies reach up to 160m in thickness and persist despite substantial surface lowering since 2019, indicating that drainage events only partially reduce the stored water volume.” Any thought from either observations or literature on the porosity/percent water? How much do these marshes dry out for example in response to 10 m surface lowering. Could be important as for example 80% porosity to 70% porosity is still effectively soup (0 strength), but 15% to 5% could reflect a fundamental shift in strength of the basal material.

Response:

Thanks for raising this point. Sediment cores from analogous features near the Antarctic Peninsula revealed a porosity of 32% to 55% with an average of 39% (Ó Cofaigh et al., 2005). The average thickness of the transparent zones we found is 70 m. Assuming the average porosity, roughly 28 m of the transparent zone is water and 42 m sediments. A surface lowering of 10 m due to the drainage of pure water would result in a 60 m transparent zone with a composition of 18 m water and 42 m sediment, which gives a porosity of 30%.

We updated the following text in the manuscript (new text in **bold**):

*L343–349: “Sediment cores from analogous features near the Antarctic Peninsula revealed that these transparent layers are composed of soft, highly porous (**39% on average**) till with massive diamicton, overlying stiff, low-porosity till (Ó Cofaigh et al., 2005). The similarity in seismic character suggests that the subglacial lake consists of a comparable soft, water-rich substrate. **The partial drainage since 2019 reduces the water content by ~10%, which could modestly increase basal drag and cause a slight deceleration in ice flow. This is consistent with minor ice-speed variations of <3% following subglacial lake filling or drainage (Smith et al., 2017, Hoffman et al., 2020).**”*

L369-370 “areas of lower-elevation that either consist of sedimentary basins (II, IV, and parts of V) or MSGSLs (upstream III and VII)”. Are sedimentary basins and MSGSL mutually exclusive? can you not get MSGSL in sedimentary basins and/or get MSGSL which mask sedimentary basins? Do you get sedimentary basins with no MSGSL? If this is correct it could be expanded further. My understanding from offshore data was that MSGSL were pretty ubiquitous wherever ice flows rapidly over sediment.

Response:

We agree that the wording we used gave the impression that MSGSL and sedimentary basins are mutually exclusive. That was not intended. MSGSLs can form in sedimentary basins, depending on the supply of sediments and the flow speed.

In the revised version, we write the sentence as follows:

*L391-393: “Elevated, rough-topography regions (I, III, VI, VIII) alternate with areas of lower-elevation that consist of sedimentary basins (II, IV, and parts of V) **and** MSGSLs (upstream III and VII).”*

Reviewer #2 (Remarks to the Author):

Review of Zeising et al.

Zeising et al. present detailed analysis of 344 km of vibroseismic profiles on Thwaites Glacier in West Antarctica. The dataset collected characterises basal properties and subglacial hydrology across an impressively-large and scientifically-important study area. The results will be key to guiding and refining future modelling efforts that aim to forecast dynamic mass loss from Thwaites. The analysis of crag and tail landforms, which play an important role in resisting ice flow, is a key aspect of this paper which could be emphasised further, perhaps with an additional annotated figure. The methods appear robust, the figures are of a high quality and support the arguments and conclusions made, and the writing is clear. This paper changes our understanding of the nature of the basal environment of Thwaites by showing how basal properties vary over a scale of 100s of kilometers both longitudinally and transverse to ice flow, and importantly across the shear margin of Thwaites. These properties are key to understanding glacier dynamics. Figure referencing in the main text could be increased slightly to help the reader navigate the paper, see specific comments below. I have only one major comment relating to the use of the term 'rheology' to describe hard beds, and a number of minor comments, both detailed below.

Samuel Doyle

Major Comment

Throughout the manuscript the term rheology is used to describe bed properties in a way that includes both hard bed sliding and deformation of subglacial sediments. It's not correct to refer to the rheology of hard bed sliding as hard beds do not flow. While it's correct to use the term for subglacial sediment deformation this usage also neglects ice-sediment decoupling which can dominate basal motion at least in some locations (e.g. Iverson et al., 1995). I think all that is required here is to use a different term to describe the nature of the bed in relation to basal motion.

Iverson, N. R., Hanson, B., Hooke, R. L., & Jansson, P. (1995). Flow mechanism of glaciers on soft beds. *Science*, 267(5194), 80-81.

Response:

Thanks for your comment on the use of the term "rheology". We agree that this word should not be used for hard, non-deformable beds. In the revised version, we now use "mechanical properties" as the general term.

Minor Comments

L20 – here ice discharge per annum is given for Thwaites Glacier and Pine Island Glacier combined and then it's stated that they are the largest regional contributors to global sea level rise. This raises two questions: why combine the discharge estimate for Thwaites with another glacier that wasn't studied? Second, why state that they are the largest contributors in this region? Few readers will understand the boundaries of this (possibly undefined) region in a way that allows them to make sense of the importance of Thwaite's annual ice discharge globally. It would be better to state the global sea level contribution of Thwaites on its own and then give it as a percentage of total global ice mass loss.

Response:

Thanks for your comment. We agree and changed the sentence in the revised version to: L20–22: *"Since 2002, the Thwaites drainage basin has lost mass at an average rate of 60.5 ± 9.2 Gt/a on average, corresponding to a global mean sea-level rise of 0.17 ± 0.03 mm/a (or 5%), making it the largest Antarctic contributor to sea-level rise (Groh & Horwath, 2021, Horwath et al., 2025)."*

Groh, A., & Horwath, M. (2021). Antarctic Ice Mass Change Products from GRACE/GRACE-FO Using Tailored Sensitivity Kernels. *Remote Sens.*, 13(9), 1736. doi:10.3390/rs13091736

Horwath, M., Döhne, T. & Knöfel, C. Gravimetric Mass Change, Antarctic Ice Sheet Project, ESA Climate Change Initiative [data set] (2025). URL https://data1.geo.tu-dresden.de/ais_gmb/. Last access: 05 December 2025.

L89 – Is GR defined earlier?

Response:

Yes, it is defined in the introduction (L45).

L89 – I presume all elevations are relative to the WGS-84 ellipsoid as defined later. Why weren't orthometric heights used? Is there a major difference between the geoid and WGS-84 ellipsoid in the study area?

Response:

We used ellipsoid heights because both our GNSS observations and the external datasets that we used (Bedmap3, BedMachine v3) are referenced to the ellipsoid. The geoid height in the study area varies between -30 and -33 m. Given that the elevation ranges are between 1000 and 4000 m, applying a geoid correction would change absolute values by <1 to 3% and would not affect any gradients or interpretations. Converting to orthometric heights would therefore introduce only a minor offset with no impact on the analysis or the figures.

L100 – ‘abruptly over some 10 km’ –over 10 km doesn’t seem very abrupt but perhaps this is valid over the length scale of the seismic line. In any case, avoid the informal ‘some’ as this would usually be used to refer to some 10s of km, which isn’t abrupt.

Response:

We agree that a multiple of 10 km is not abrupt. The main increase in bed elevation occurs within 10 km (~km 45 to 55). Thus, we will remove “some” in the sentence.

Figure 1 – capitalise ‘Surf. Vel. on (a).

Response:

We created all figures with lowercase axis labels, therefore we prefer not to change it to capitalized labels.

L113 – UTR has previously been defined so doesn’t need to be defined again here.

Response:

Thanks for noticing! We removed the new definition.

L113 – Add “at” to “located at km 115-145”

Response:

Thanks! We added “at”.

L128 – The moat on Figure 5 at km 10 looks deeper than 12 m. It would be good to expand on moats and crag and tails here or in the introduction so the reader understands what they are prior to their mention in the discussion. Moats aren’t a well known glacial landform as far as I’m aware so a brief introduction is warranted.

Response:

Thanks for noticing! This was a typo. We wanted to write 120 m. We corrected this in the revised version.

In the results section, we describe the crag-and-tail landform as follows:

L128–130: “Several of these features are probably crag-and-tail landforms comprising a steep, approximately 100 m high stoss side (crag), followed by a lee-side tail extending for several kilometres.”

In the revised version, we introduce moats as follows:

L133–134: “These moats are narrow, elongated depressions, formed by focused basal water flow or enhanced erosion adjacent to harder bed obstacles.”

L138 – Give figure reference for stratification.

Response:

We added a reference to Supplementary Figure 1.

Figure 2 caption – expand on what this unpublished dataset is in the manuscript.

Response:

This (so far) unpublished data set is a seismic reflection survey (using explosives) from summer 2023–24 (Agnew et al., 2025 (EGU Abstract)).

In the revised version, we added information:

*“[...] and an unpublished **seismic reflection survey from summer 2023–24 in region I** (Agnew et al., 2025).”*

Agnew, R., Brisbourne, A., Anandakrishnan, S., Muto, A., Borthwick, L., Willet, A., and Melton, S. and the ITGC GHOST Team: Seismic reflection surveys at GHOST Ridge, Thwaites Glacier, EGU General Assembly 2025, Vienna, Austria, 27 Apr–2 May 2025, EGU25-12160, <https://doi.org/10.5194/egusphere-egu25-12160>, 2025.

L169 – state which seismic line with an appropriate label (e.g. E-W or transverse).

Response:

We added “across-flow profile” to clarify to which profile this sentence refers to.

L160 – Give figure reference to Figures 5 and S5. Same applies to similar sections below.

Response:

We added figure references here and below in this section.

L170 – can the subglacial lake locations be marked on Figure S5?

Response:

The Figure S5 (now S6) shows the lake outline (white dotted line in **a**, described in the figure legend) and labels of the transparent zones in **b**. In **a**, we added the label “Thw₁₂₄” similarly to Fig. 5.

L169 “covering” -> “cover”

Response:

Thanks, we corrected this in the revised version.

L185 – add “elevation” to “bed elevation”.

Response:

Thanks! We corrected this in the revised version.

L217 – where specifically are you referring to outside of Thwaites Glacier – give km reference.

Response:

With “outside of Thwaites Glacier”, we refer to km 115 to 127. We added this to the sentence.

L224 – Bow tie structures are mentioned but never explained or expanded on. Can this be addressed briefly?

Response:

Thanks for your comment. We agree that mentioning “bow-tie structure” would require further explanations. However, this part of the sentence does not include essential information. We therefore removed it.

L231 – the last sentence in this paragraph isn’t a complete sentence and needs revising and the point made needs expanding on to be worthwhile including.

Response:

Thanks for noticing! We improve the sentence as follows:

L243–244: “Englacial reflections may extend to greater heights, but they were obscured by the direct and diving wave.”

L238 – The sentence starting ‘seismic measurements are essential’ can be omitted as this argument was established in the introduction.

Response:

We agreed and removed the sentence.

L243 – Go further in this first sentence and state that the topography and roughness constrained by the results presented improves on previous datasets– tell the reader here what you go on to tell them later in the paragraph.

Response:

Thanks! In the revised version, we wrote the first sentence of the paragraph as follows:

L252–254: “*The vibroseismic profile in along-flow direction **provides improved constraints on bed topography and basal roughness relative to existing regional datasets, revealing pronounced large-scale variability.***”

L243 – Consider whether the phrase ‘the bed is characterised by significant roughness and large bed slopes’ is excessively verbose in comparison to something like “the bed is rough and steeply sloping.”

Response:

Thanks! We agreed and followed your suggestion.

L247 – qualify the new sentence with “In these simulations,”

Response:

Thanks, we followed your suggestion!

L258 – Remove repeated last sentence.

Response:

Thanks for finding this mistake. We removed the first sentence.

L274 and L277 – omit ‘also’

Response:

Thanks!

L289 – This section on crag and tails is excellent and a key aspect of the paper. Can you give a figure reference to an example in the seismogram? Or maybe even create a new figure of a crag and tail example with more detail and annotations.

Response:

We agreed and added a new figure to the Supplementary Information (Supplementary Figure 3) and referenced this figure in the Results section and in the discussion on crag-and-tails. The figure contains a zoom on a crag-and-tail with annotations.

In the previous version of the manuscript, we interpreted the parallel reflections in the tail as stratification. However, these are most likely off-nadir reflections. We therefore removed the sentence from the discussion.

L298 – Figure reference and km reference here would help the reader.

Response:

Thanks! We added figure references and km:

*L316–319: “The along-flow vibroseismic profile reveals distinct crag-and-tails just upstream of the ridges (**km 137 to 143, Supplementary Figure 2, 3**), where airborne radar-derived bed topography shows several parallel features aligned with ice flow.”*

L304 – New paragraph here.

Response:

Thanks!

L314 – plural ‘water bodies’ – singular lake. Choose one of these?

Response:

We agree that it can be confusing to write about water bodies although the seismic cross-section does show several water bodies in the lake. However, as long as these water bodies are closely hydrologically connected, we can define it as a single lake.

Thus, we improved the wording in the sentences:

L332–336: “Our vibroseismic observations provide the initial evidence suggesting that the subglacial lake Thw₁₂₄ is composed of highly porous, water-saturated sediments. [...] The lake reaches up to 160 m in thickness and persists despite substantial surface lowering since 2019, indicating that drainage events only partially reduced the stored water volume.”

L315 – use past tense ‘reduce’

Response:

Thanks!

L320 – lower case “sonar”

Response:

Thanks!

L321 – The sentence starting Evans et al. appears to directly follow on from the previous. If so, link these explicitly with a semi-colon.

Response:

Yes, the second sentence follows the first one. We combined these with a semicolon in the revised version.

L336 – add “shear” to “shear margin”.

Response:

We added "shear". Thanks!

L374 – Make “structure” plural and change “was” to “were”.

Response:

Thanks!

L379 – this sentence says you observed water in both depressions and hills but it would be useful to know whether the former was more common.

Response:

We improved the sentences as follows:

*L404–407: “For the first time, we directly image the subglacial hydrology across several hundred kilometres, revealing that water-filled sediments **occur predominantly** in bedrock depressions, with **smaller and more isolated patches** also present along and on top of topographic highs.”*

L395 – citation for statistical extrapolations.

Response:

We added the following citation as an example of such extrapolation:

Shackleton, C., Matsuoka, K., Moholdt, G., Van Liefferinge, B., & Paden, J. (2023). Stochastic simulations of bed topography constrain geothermal heat flow and subglacial drainage near Dome Fuji, East Antarctica. *Journal of Geophysical Research: Earth Surface*, 128, e2023JF007269. <https://doi.org/10.1029/2023JF007269>

L419 – ‘do’ to ‘to’

Response:

Corrected, thanks!

L462-466 – correct reverse quotation marks.

Response:

Thanks! We corrected this!

Remove ‘as’ on L465.

Response:

Done!

L471 – give a figure reference for the surface elevation data in this section.

Response:

We now referenced Fig. 5b here, which shows the surface elevation changes.

L649 – give link to data repository.

Response:

So far, only a temporal link to the data repository exists:

<https://doi.pangaea.de/10.1594/PANGAEA.987704>

The data files are under embargo and can not be accessed yet.

The final doi will be: <https://doi.org/10.1594/PANGAEA.987704>

We added this doi to the manuscript.

Supplementary Material

L10 – mark new paragraph with either a vertical space between paragraphs or an indent.

Response:

Done!

L37 – what geological features are being referred to?

Response:

With “geological features”, we meant: sedimentary basins, tails of crag-and-tails and transparent zones

We improved the sentence as follows:

SM L37–38: *“The interval velocities of sedimentary basins, tails of crag-and-tails and transparent zones were determined based on the Dix-Dürbaum-Krey equation.”*

Reviewer #3 (Remarks to the Author):

This manuscript presents an extensive and technically sophisticated vibroseismic investigation of the Thwaites Glacier bed, offering new insights into basal topography, rheology, and hydrology. The study's scope—344 km of seismic profiling—is impressive and contributes valuable empirical data to one of the most critical regions for ice-sheet stability and sea-level projections. The integration of amplitude, velocity, and impedance analyses, combined with cross-referencing to radar and altimetry datasets, demonstrates strong methodological rigor. It is also great to see that the data are uploaded to World Data Center PANGAEA.

However, while the work is clearly of high potential significance, several areas could be strengthened to enhance clarity, reproducibility, and scientific impact:

1. The introduction is well contextualized within the literature on Thwaites Glacier, but the novelty of this study is not fully articulated. The paper often reiterates that current models lack the complexity revealed by the new seismic data, yet it could more explicitly state what specific knowledge gaps this study closes compared to prior seismic work (e.g., Muto et al. 2019; Clyne et al. 2020). A concise statement of unique contributions—whether in spatial coverage, resolution, or methodological innovation—would strengthen the manuscript's conceptual foundation.

Response:

Thank you for your comment. We agree that the last paragraph of the introduction focused on how seismic profiles can contribute to improving models, rather than addressing the knowledge gaps that this study tries to close.

In the revised version, we improved the last paragraph of the introduction by highlighting that the seismic profiles have a substantially larger continuous extent than the previous seismic studies (from Muto et al. 2019 and Clyne et al. 2020), while maintaining high spatial resolution and even improving signal-to-noise ratio by higher fold. This allowed us to resolve basal properties at scales that were not previously observed at Thwaites Glacier.

*L74–83: “Here, we present 344 km of vibroseismic profiles collected along and across Thwaites Glacier. **This dataset provides substantially greater spatial coverage and a higher signal-to-noise ratio than previous studies, while maintaining high horizontal resolution.** We combine amplitude, velocity, and structure analysis of seismic reflections to characterize basal topography, subglacial materials, and hydrological features, **thereby resolving basal heterogeneity at scales not previously observed at Thwaites Glacier or elsewhere underneath grounded ice in Antarctica.** By evaluating acoustic impedance and interpreting it in terms of spatial variable basal drag, our results provide critical constraints for ice sheet models. Such constraints are particularly important for improving projections of future mass loss from one of Antarctica’s most vulnerable outlet glaciers, where basal conditions strongly influence ice acceleration and grounding-line retreat.”*

2. Although the amplitude and impedance analyses are detailed, the manuscript provides limited discussion of uncertainty quantification and its influence on geological interpretation. Given the heterogeneous basal conditions, explicit error ranges for impedance-derived classifications (“hard,” “soft,” “water-saturated”) are essential. For instance, slope effects, off-nadir reflections, and local anisotropy could bias impedance contrasts; these sources of error should be more systematically evaluated. Moreover, some interpretations (e.g., identifying “water-saturated sediments” vs. “unconsolidated till”) rely heavily on qualitative comparison to prior work and could benefit from more quantitative justification.

Response:

Thank you for highlighting the need for a clearer discussion of uncertainty and its implications for geological interpretation. We have expanded the manuscript to address these additional uncertainty sources and improved the impedance-derived bed classifications.

In the previous version of the manuscript, we provide a quantitative uncertainty assessment for the reflection coefficient and acoustic impedance, accounting for uncertainties in source amplitude, attenuation within the ice column, and the acoustic impedance of ice. The resulting error bounds are generally small and do not lead to any uncertainties regarding the quantification of mechanically hard and soft basal materials.

A more precise discrimination between specific soft-bed endmembers (e.g., dilatant till vs. unconsolidated sediments) is more uncertain. This limitation arises from several factors. Reported acoustic impedance ranges for subglacial materials in the literature are broad and often overlapping. In principle, amplitude-versus-angle (AVA) analysis could provide additional constraints on basal rheology (Zechmann et al., 2018); however, the maximum offset of 1500 m, combined with ice thicknesses exceeding 2000 m, precludes a reliable AVA analysis in our dataset.

Zechmann, J. M. et al. Active seismic studies in valley glacier settings: strategies and limitations. *Journal of Glaciology* 64, 796–810 (2018)

Additional potential sources of uncertainties are basal slope effects, off-nadir reflections and anisotropy in the ice. For most of the profiles, basal slopes are sufficiently small (mean absolute slope $\approx 3\%$) such that slope-induced amplitude distortions are expected to be minor. Steeper slopes primarily occur on the stoss sides of crag-and-tail bedforms, where impedance values fall within the range of consolidated or lithified sediments. For such materials, the AVA response is known to be relatively flat for incidence angles up to approximately $20\text{--}30^\circ$, suggesting limited sensitivity to moderate slope variations. Off-nadir reflections are present in parts of the data but predominantly occur below the first basal return. Assuming that only narrow off-nadir reflections potentially occur as a first basal return, the impact on the impedance estimates is minor. Anisotropic ice mainly influences the propagation velocity, which we treated carefully.

While we cannot quantitatively isolate the contribution of each of these effects to the total uncertainty, we do not expect them to significantly alter the interpretation of basal mechanical regimes.

In the revised version, we now mention the additional sources of uncertainty:

L484–488: “Beyond the uncertainties explicitly accounted for in the impedance analysis, additional processes may influence the derived acoustic impedance, including basal slope effects, off-nadir reflections, and debris entrained within the basal ice. Based on the generally low bed slopes and the characteristics of the observed basal reflections, we expect these effects to be of secondary importance and unlikely to significantly bias the first-order impedance-based bed classification.”

We agree that the distinction between “water-saturated sediments” and “unconsolidated sediments” (or “dilatant till”) was not sufficiently clear in the original manuscript. In the revised version, we have clarified both the terminology and the underlying rationale. We now use the term “water-saturated sediments”, where we want to highlight the presence of water indicated by acoustic impedance or interval velocities that fall below the range typically reported for dilatant till (or unconsolidated sediments) in the literature.

In the revised version, we write:

L173–176: “[...] we observe consistent indicators of basal water-sediment mixtures beneath the ice. These include predominantly negative reflection coefficients and lower acoustic impedance values than expected for dilatant till, pointing to a soft basal layer with varying porosity/effective pressure, possibly with the presence of the presence of free water.”

This classification does not imply a fundamentally different sediment type; rather, it reflects a higher effective porosity within the dilatant till.

We have incorporated “dilatant till” into the manuscript text and figures and revised the classification scheme in the Methods section:

L492–502: “In addition, we classify basal conditions into three categories based on inferred resistance to deformation, using acoustic impedance as a proxy for basal mechanical behaviour (Supplementary Table 3). We distinguish between a “hard bed”, where basal motion is dominated by sliding at the ice–bed interface, and a “soft bed”, where motion is primarily due to deformation of subglacial till. A “hard bed” consists of consolidated and lithified sediments with an acoustic impedance $Z_b > 3.24 \times 10^6 \text{ kg m}^{-2} \text{ s}^{-1}$. In contrast, a “soft bed” consists of dilatant till with a lower impedance than unconsolidated, stiff sediments ($Z_b < 2.72 \times 10^6 \text{ kg m}^{-2} \text{ s}^{-1}$). We define the transition between soft and hard beds as an “intermediate bed”, with acoustic impedance of $2.72 \times 10^6 < Z_b < 3.24 \times 10^6 \text{ kg m}^{-2} \text{ s}^{-1}$. For this class, the limited sensitivity of small-offset seismic acquisitions precludes a unique determination of the dominant basal deformation mechanism.”

In the main text, we included the distribution of these categories as found within the two profiles, which we visualise in a new figure:

L281–283: “At Thwaites Glacier, our seismic measurements and derived acoustic impedance reveal a mixed bed, comprising approximately 50% hard bed, 43% soft bed, and 7% intermediate bed (Fig. 7, Supplementary Figure 11).”

3. The discussion emphasizes that current models underrepresent basal complexity but stops short of connecting these findings to specific modelling frameworks or parameters. How might the new impedance-derived rheology maps alter basal drag formulations or grounding-line stability simulations? A schematic or conceptual model linking seismic observations to potential model parameterization would make the results more actionable for ice-sheet modelers.

Response:

The goal of this manuscript is to present new observations of subglacial properties that can be used to constrain these modeling frameworks and parameters. Recently developed and applied methods show how sliding law parameters can be directly inferred from the acoustic impedance (Hank et al., 2025, Preprint). In the revised version, we now included a short paragraph about the outlook of how the spatial varying basal drag can be derived from our acoustic impedance values.

L287–292: “Recently developed methods allow the inference of sliding laws from seismic-derived acoustic impedance (Hank et al., 2025). Consequently, our dataset and observations have the potential to inform the sliding laws over an unprecedented area and in high detail, which will improve predictive models of Thwaites Glacier’s dynamics and its contribution to sea-level rise. However, determining the sliding laws that are most appropriate for our soft bed/hard/bed classification is beyond the scope of this study.”

Hank, K., Arthern, R. J., Williams, C. R., Brisbourne, A. M., Smith, A. M., Smith, J. A., Wåhlin, A., and Anandakrishnan, S.: The Antarctic Ice Sheet sliding law inferred from seismic observations, EGU sphere [preprint], <https://doi.org/10.5194/egusphere-2025-764>, 2025.

4. The numerous figures are technically rich but sometimes dense. Readers may struggle to extract key spatial relationships without frequent cross-referencing between figures 1–6. Simplified summary figures or conceptual diagrams synthesizing major bed types (hard rock, sediment basins, water pockets) along both profiles would improve readability.

Response:

Thanks for your idea! We understand the need for a figure that summarizes the key aspects in a simpler way. We created a new 3D figure (Fig. 7) that shows part of both vibroseismic profiles and visualises the key features and the identified bed types: hard, intermediate and soft. Additionally, we added a figure to the Supplementary Information (Supplementary Figure 11) showing the bed classification of both profiles and a video illustration showing the both seismic profiles in context of Thwaites.

Fig. 7: Three-dimensional visualisation of vibroseismic profiles and derived basal properties. View of the two crossing vibroseismic profiles together with the classification of basal conditions into hard, intermediate, and soft beds (colour). Key subglacial and englacial features identified from the seismic interpretation are annotated. The black lines at the glacier surface show the outline of lake Thw₁₂₄ as derived from surface elevation changes. The background shows the bed elevation (WGS 84) from Bedmap3. The image is vertically exaggerated by a factor of 10, and ice flow is from south to north.

Supplementary Figure 11: Bed classification of the along-flow and across-flow profiles. **a, c** Kirchhoff migrated vibroseismic profiles in along-flow (a) and across-flow (c) direction with a vertical exaggeration $VE = 10$. **b, d** Classification of basal conditions into hard ($Z_b > 3.24 \times 10^6 \text{ kg m}^{-2} \text{ s}^{-1}$), intermediate ($2.72 \times 10^6 < Z_b < 3.24 \times 10^6 \text{ kg m}^{-2} \text{ s}^{-1}$), and soft beds ($Z_b < 2.72 \times 10^6 \text{ kg m}^{-2} \text{ s}^{-1}$).

5. The section on englacial reflections introduces an intriguing discussion of ice fabric anisotropy but remains speculative. The link between seismic reflections, fabric orientation, and shear-margin rheology should either be supported with more direct data or framed more cautiously. Distinguishing between englacial scattering, layer reflectivity, and genuine crystal orientation effects requires stronger methodological justification. In particular, it will be helpful to have a representative waveform figure to clearly show the englacial reflections along with other key phases, such as in Fig.S12.

Response:

Thank you for this comment. The primary aim of our study is to characterize basal properties beneath Thwaites Glacier. We also included the topic of ice crystal anisotropy based on the propagation velocity analysis (similar to anisotropy in radar data (Hills et al. 2025)) and the observed englacial reflections to provide a complete description of the data. However, we have refrained from interpreting their origin in detail, as this would go beyond the scope of this study. Instead, we relate the findings of englacial reflections to similar observations reported in previous studies (e.g. Horgan et al., 2011, Hofstede et al., 2013) in which englacial reflections were linked to fabric.

Hills, B.H., Young, T.J., Lilien, D.A., Babcock, E., Bienert, N., Blankenship, D., et al. (2025). Radar polarimetry in glaciology: Theory, measurement techniques, and scientific applications for investigating the anisotropy of ice masses. *Reviews of Geophysics*, 63, e2024RG000842. <https://doi.org/10.1029/2024RG000842>

Here, we highlight that further work is needed to better interpret (our) observations and to better understand the ice rheology.

We agree that the conclusions gave the impression that the observed englacial reflection were directly linked to changes in fabric orientation. In the revised version, we have highlighted the linkage between englacial reflections and abrupt changes in crystal orientation fabric in the discussion and rephrased the conclusions:

*L415–418: “Seismic wave speed variability across the glacier indicates a strong orientation of ice crystal fabric, particularly near the shear margins, which **can** affect the ice deformation. **Widespread englacial reflections above the bed may reflect variations in ice crystal fabric, but their analysis goes beyond the scope of this study.**”*

Hard rocks and deep wetlands beneath Thwaites Glacier in Antarctica

Point-by-point response

Dear Editor, dear Reviewers,

We would like to thank you again for your efforts to improve the quality of our manuscript. In the revised version, we corrected the missing minus sign, and followed the editorial comments and requests.

Thank you once again for your time and expertise.

Kind regards,
Ole Zeising and co-authors

Reviewer #1 (Remarks to the Author):

Second review of "Hard rocks and deep wetlands: characterizing the bed of Thwaites Glacier, Antarctica, from vibroseismic measurements".

This paper presents new seismic reflection data from the central part of Thwaites Glacier. The authors have responded well to my previous comments and I am happy to see the paper move forward to publication.

One minor note – Line 119 "LTR has the highest elevation of 1050m, surpassing its surroundings by approximately 350 to 400m". I think this should be an elevation of -1050m.

Response:

Thanks for noticing. We corrected this in the revised version.

Reviewer #2 (Remarks to the Author):

My previous comments have been satisfactorily addressed. I have no further comments.

Response:

Thanks!

Reviewer #3 (Remarks to the Author):

The revised manuscript thoroughly address my comments, I appreciate the new Fig.7. I have no further concern to the manuscript and ready to sign off. Congradulations to the authors for a nice piece of work!

Response:

Thanks!